# BIDIRECTIONAL-REACHABLE HIERARCHICAL RL WITH MUTUALLY RESPONSIVE POLICIES

## ABSTRACT

Hierarchical reinforcement learning (HRL) addresses complex long-horizon tasks by skillfully decomposing them into subgoals. Therefore, the effectiveness of HRL is greatly influenced by subgoal reachability. Typical HRL methods only consider subgoal reachability from the unilateral level, where a dominant level enforces compliance to the subordinate level. However, we observe that when the dominant level becomes trapped in local exploration or generates unattainable subgoals, the subordinate level is negatively affected and cannot follow the dominant level's actions. This can potentially make both levels stuck in local optima, ultimately hindering subsequent subgoal reachability. Allowing real-time bilateral information sharing and error correction would be a natural cure for this issue, which motivates us to propose a mutual response mechanism. Based on this, we propose the Bidirectional-reachable Hierarchical Policy Optimization (BrHPO)—a simple yet effective algorithm that also enjoys computation efficiency. Experiment results on a variety of long-horizon tasks showcase that BrHPO outperforms other state-of-the-art HRL baselines, coupled with a significantly higher exploration efficiency.

## 1 INTRODUCTION

Reinforcement learning (RL) has demonstrated impressive capabilities in decision-making scenarios, ranging from achieving superhuman performance in games (Mnih et al., 2015; Lample & Chaplot, 2017; Silver et al., 2018), developing complex skills in robotics (Levine et al., 2016; Schulman et al., 2015) and enabling smart policies in autonomous driving (Jaritz et al., 2018; Kiran et al., 2021; Cao et al., 2023). Most of these accomplishments are attributed to single-level methods (Sutton & Barto, 2018), which learn a flat policy by trial and error without extra task decomposition or subgoal guidance. While single-level methods excel at short-horizon tasks involving inherently atomic behaviors (Levy et al., 2018; Nachum et al., 2018b; Pateria et al., 2021b), they often struggle to optimize effectively in long-horizon complex tasks that require multi-stage reasoning or sparse reward signals. To address this challenge, hierarchical reinforcement learning (HRL) has been proposed, aiming to decompose complex tasks into a hierarchy of subtasks or skills (Kulkarni et al., 2016; Bacon et al., 2017; Vezhnevets et al., 2017). By exploiting subtask structure and acquiring reusable skills, HRL empowers agents to efficiently solve long-horizon tasks.

Subgoal-based HRL methods, a prominent paradigm in HRL, partition complex tasks into simpler subtasks by strategically selecting subgoals to guide exploration (Vezhnevets et al., 2017; Nachum et al., 2018b). Subgoal reachability is crucial in evaluating how effectively the low-level policies' exploration trajectory aligns with the high-level policy's subgoal, ultimately determining task performance (Vezhnevets et al., 2017; Zhang et al., 2020). However, existing approaches for improving subgoal reachability predominantly focus on one level of the hierarchical policy, imposing dominance on the other level. This can be categorized as either low-level dominance or high-level dominance (Nachum et al., 2018b; Zhang et al., 2020; Andrychowicz et al., 2017; Chane-Sane et al., 2021; Eysenbach et al., 2019; Jurgenson et al., 2020). Low-level dominance (Figure 1a) refers to the accommodation of low-level passive exploratory behaviour by the high-level policy, causing the agent to get stuck near the starting position. On the other hand, high-level dominance (Figure 1b) may result in unattainable subgoals, causing repeated failure and sparse learning signals for the low-level policy. To assess these methods, we applied them to two HRL benchmarks, AntMaze and AntPush, and generated state-subgoal trajectories for visualization. The results reveal that the former methods exhibit lower exploration efficiency as the high level must generate distant subgoals to guide the low

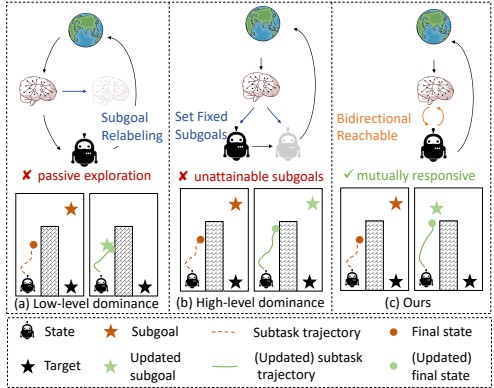
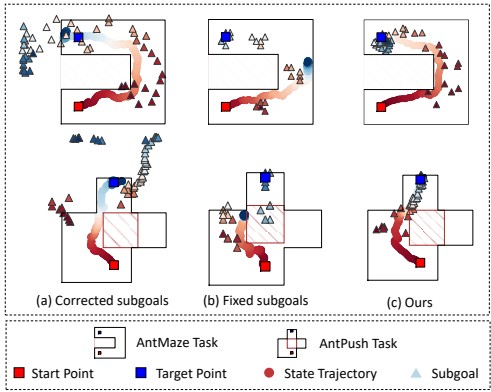

Figure 1: A motivating example of our proposed BrHPO. The earth, brain, and robot symbols stand for the environment, high-level policy, and low-level policy, respectively.

Figure 2: The state-subgoal trajectory comparison of HIRO (a), RIS (b) and BrHPO (c). BrHPO can improve the alignment between states and subgoals, thus benefitting overall performance.

level (Figure 2a), while the latter methods may create unattainable subgoals, resulting in the low-level policy's inability to track them (Figure 2b).

Enforcing subgoal reachability through unidirectional communication between the two levels has limitations in overall performance improvement. A bidirectional reachability approach, illustrated in Figure 1c, holds the potential to be more effective in HRL. From an optimization perspective, bidirectional reachability provides two key benefits: 1) the high-level policy can generate subgoals that strike a balance between incentive and accessibility, and 2) the low-level policy can take more effective actions that drive subtask trajectories closer to the subgoal. Despite its potential advantages, bidirectional subgoal reachability has not been extensively studied in previous research, and its effectiveness in enhancing HRL performance requires further investigation. We explore the theoretical benefits of bidirectional insights, and empirically demonstrate its effectiveness through visualizing the alignment between states and subgoals in Figure 2 and our ablation studies.

This paper aims to investigate the potential of bidirectional subgoal reachability in improving subgoal-based HRL performance, both theoretically and empirically. Specially, we propose a joint value function and then derive a performance difference bound for hierarchical policy optimization. The analysis suggests that enhancing subgoal reachability, from the mutual response of both-level policies, can effectively benefit overall performance. Motivated by these, our main contribution is a simple yet effective algorithm, Bidirectional-reachable Hierarchical Policy Optimization (BrHPO) which incorporates a mutual response mechanism to efficiently compute subgoal reachability and integrate it into hierarchical policy optimization. Through empirical evaluation, we demonstrate that BrHPO achieves promising asymptotic performance and exhibits superior training efficiency compared to state-of-the-art HRL methods. Additionally, we investigate different variants of BrHPO to showcase the effectiveness and robustness of the proposed mechanism.

## 2 PRELIMINARIES

We consider an infinite-horizon discounted Markov Decision Process (MDP) with state space $\mathcal{S}$, action space $\mathcal{A}$, goal/subgoal space $\mathcal{G}$, unknown transition probability $P^a_{s,s'} : \mathcal{S} \times \mathcal{A} \times \mathcal{S} \to [0,1]$, reward function $r : \mathcal{S} \times \mathcal{A} \times \mathcal{G} \to \mathbb{R}$, and discounted factor $\gamma \in (0,1)$.

**Framework of subgoal-based HRL.** Subgoal-based HRL, also called Feudal HRL (Dayan & Hinton, 1992; Vezhnevets et al., 2017), comprises two hierarchies: a high-level policy generating subgoals, and a low-level policy pursuing subgoals in each subtask. Assume that each subtask contains a fixed length of $k$ timesteps. At the beginning of the $i$-th subtask where $i \in \mathbb{N}$, the high-level policy $\pi_h$ observes state $s_{ik}$ and then outputs a subgoal $g_{(i+1)k} \sim \pi_h(\cdot|s_{ik}) \in \mathcal{G}$. Then, in each subtask, the low-level policy $\pi_l$ performs actions conditioned on the subgoal and the current state, $a_{ik+j} \sim \pi_l(\cdot|s_{ik+j}, g_{(i+1)k}) \in \mathcal{A}$, where $j \in [0, k-1]$ is a pedometer in a subtask. With the

guidance from the subgoal, the state-subgoal-action trajectory in the $i$-th subtask comes out to be

$$\tau_i^{\pi_h,\pi_l} \triangleq \left\{ (s_{ik+j}) | s_{ik}, g_{(i+1)k} \sim \pi_h(\cdot | s_{ik}, \hat{g}), a_{ik+j} \sim \pi_l(\cdot | s_{ik+j}, g_{(i+1)k}) \right\}_{j=0}^{k-1}, \tag{2.1}$$

and the whole task trajectory forms by stitching all subtask trajectories as $\tau = \cup_{i=0}^{\infty} (\tau_i^{\pi_h,\pi_l})$.

Following prior methods (Andrychowicz et al., 2017; Nachum et al., 2018b; Zhang et al., 2020), we optimize $\pi_h$ based on the high-level reward $r_h$, defined as the environment reward feedback summed over a subtask

$$r_h(\tau_i^{\pi_h,\pi_l}) = r_h(s_{ik}, g_{(i+1)k}) = \sum_{j=0}^{k-1} r(s_{ik+j}, a_{ik+j}, \hat{g}), \tag{2.2}$$

and the intrinsic reward for the low-level policy $\pi_l$ is

$$r_l(s_{ik+j}, a_{ik+j}, g_{(i+1)k}) = -\mathcal{D}(\psi(s_{ik+j+1}), g_{(i+1)k}). \tag{2.3}$$

where $\psi : \mathcal{S} \mapsto \mathcal{G}$ is a pre-defined state-to-goal mapping function and $\mathcal{D} : \mathcal{G} \times \mathcal{G} \to \mathbb{R}_{\geq 0}$ is a chosen binary or continuous distance measurement.

**Unilateral subgoal reachability in hierarchical policy optimization.** To optimize both $\pi_h$ and $\pi_l$, previous methods would establish the separated value functions with the initial state $s_0 \in d_0(s)$,

$$V^{\pi_h}(s_0) = \sum_i^{\infty} \gamma^i \mathbb{E}_{s \sim \mathbb{P}_{ik}^{\pi_l,g}(\cdot | s_0), g \sim \pi_h(\cdot | s, \hat{g})} \left[ r_h(s_{ik}, g_{(i+1)k}) \right], \tag{2.4}$$

$$V^{\pi_l}(s_0) = \sum_t^{\infty} \gamma^t \mathbb{E}_{s, a \sim \mathbb{P}_t^{\pi_l,g}(\cdot, \cdot | s_0), g \sim \pi_h(\cdot | s, \hat{g})} \left[ r_l(s_t, a_t, g) \right], \tag{2.5}$$

where $\mathbb{P}_t^{\pi_l,g}(s, a | s_0)$ represents the probability of $\pi_l(\cdot | s, g)$ reaching the state-action pair $(s, a)$ at time step $t$ and $\mathbb{P}_t^{\pi_l,g}(s | s_0) = \sum_a \mathbb{P}_t^{\pi_l,g}(s, a | s_0)$ as the state reaching probability.

Directly maximizing the high- and low-level value functions in isolation through off-policy training methods, such as Vanilla HRL (Kulkarni et al., 2016; Wöhlke et al., 2021), would result in performance oscillation or task failure due to the dynamically varying and inter-nested nature of the hierarchical policies. To mitigate these issues, recent studies (Pateria et al., 2021b) have proposed incorporating subgoal reachability to bridge the high- and low-level optimization. However, relying solely on unilateral subgoal reachability can lead to the over-optimization of the value function in the dominant layer, while imposing additional constraints (Zhang et al., 2020) or changing training experience (Nachum et al., 2018b) for the follower level's optimization, both of which can lead to increased computational complexity and passive exploration.

## 3 INVESTIGATION ON BIDIRECTIONAL SUBGOAL REACHABILITY IN HRL

In this section, we investigate the theoretical benefits of bidirectional subgoal reachability compared with the unilateral one. Specifically, we first construct a joint value function to estimate the performance of hierarchical policies. Following up, a performance difference bound of subgoal-based HRL is derived, which supplies the design insight of the mutual response mechanism.

### 3.1 JOINT VALUE FUNCTION FOR SUBGOAL-BASED HRL

To evaluate the overall performance of subgoal-based HRL, we construct a joint value function by calculating the discounted sum of step-wise rewards accumulated along the trajectory generated by both the high- and low-level policies, as presented below:

**Definition 3.1** (Joint Value Function of Hierarchical Policies). *The long-term cumulative return* $V^{\pi_h,\pi_l}(s_0)$ *of the subgoal-based HRL in the real environment can be defined as,*

$$V^{\pi_h,\pi_l}(s_0) = \sum_t^{\infty} \gamma^t \mathbb{E}_{s, a \sim \mathbb{P}_t^{\pi_l,g}(\cdot, \cdot | s_0), g \sim \pi_h(\cdot | s)} \left[ r(s_t, a_t, \hat{g}) \right]. \tag{3.1}$$

Then, we turn to derive a refined joint value function, by decomposing it via the summation of subtasks (refer to the blue part),

$$V^{\pi_h, \pi_l}(s_0) = \sum_{i=0}^{\infty} \mathbb{E}_{g \sim \pi_h(\cdot|s)} \left[ \gamma^{ik} \left( \sum_{j=0}^{k-1} \gamma^j \mathbb{E}_{s,a \sim \mathbb{P}_{ik+j}^{\pi_l,g}(\cdot,\cdot|s_0)} r(s_{ik+j}, a_{ik+j}, \hat{g}) \right) \right]. \qquad (3.2)$$

With the joint value function established, we could use a single value function to assess the overall performance of bi-level policies, which enables us to easily construct a performance difference bound.

## 3.2 THEORETICAL INSIGHTS FROM PERFORMANCE DIFFERENCE BOUND

To investigate the optimality of the policies, we derive a performance difference bound between an induced optimal hierarchical policy $\Pi^* = \{\pi_h^*, \pi_l^*\}$ and a learned one $\Pi = \{\pi_h, \pi_l\}$, which can be formulated as $V^{\Pi^*}(s) - V^{\Pi}(s) \leq C$. Hence, the learned hierarchical policy $\Pi$ can be optimized by minimizing the upper bound $C$.

**Theorem 3.2** (Sub-optimal performance difference bound of HRL). *The performance difference bound $C$ between the induced optimal hierarchical policies $\Pi^*$ and the learned one $\Pi$ can be*

$$C(\pi_h, \pi_l) = \frac{2r_{max}}{(1-\gamma)^2} \left[ \underbrace{(1+\gamma)\mathbb{E}_{g \sim \pi_h} \left( 1 + \frac{\pi_h^*}{\pi_h} \right) \epsilon_{\pi_l^*, \pi_l}^g}_{\text{(i) hierarchical policies' inconsistency}} + \underbrace{2 \left( \mathcal{R}_{max}^{\pi_h, \pi_l} + 2\gamma^k \right)}_{\text{(ii) subgoal reachability penalty}} \right], \qquad (3.3)$$

*where $\epsilon_{\pi_l^*, \pi_l}^g$ is the distribution shift between $\pi_l^*$ and $\pi_l$, and $\mathcal{R}_{max}^{\pi_h, \pi_l}$ is the maximum subgoal reachability penalty from the learned one $\Pi$, both of which are formulated as,*

$$\epsilon_{\pi_l^*, \pi_l}^g = \max_{s \in \mathcal{S}, g \sim \pi_h} D_{TV} \left( \pi_l^*(\cdot|s,g) \| \pi_l(\cdot|s,g) \right) \quad \text{and} \quad \mathcal{R}_{max}^{\pi_h, \pi_l} = \max_{i \in \mathbb{N}} \mathcal{R}_i^{\pi_h, \pi_l}.$$

*Summary of proof.* We first divide the bound into three parts, $V^{\Pi^*}(s) - V^{\Pi}(s) = \underbrace{V^{\pi_h^*, \pi_l^*}(s_0) - V^{\pi_h^*, \pi_l}(s_0)}_{L_1} + \underbrace{V^{\pi_h^*, \pi_l}(s_0) - V^{\pi_h, \pi_l^*}(s_0)}_{L_2} + \underbrace{V^{\pi_h, \pi_l^*}(s_0) - V^{\pi_h, \pi_l}(s_0)}_{L_3}$. Then, we find the similarity of $L_1$ and $L_3$, both of which denote that under the same high-level policy ($\pi_h^*$ in $L_1$ while $\pi_h$ in $L_3$). By Performance Difference Lemma (Kakade & Langford, 2002), we have

$$L_1 + L_3 \leq \frac{2r_{max}}{(1-\gamma)^2} \mathbb{E}_{g \sim \pi_h} \left( 1 + \frac{\pi_h^*}{\pi_h} \right) \epsilon_{\pi_l^*, \pi_l}^g. \qquad (3.4)$$

To deal with the middle term $L_2$, in each subtask we define the variable $\mathcal{R}_i^{\pi_h, \pi_l}$ by

$$\mathcal{R}_i^{\pi_h, \pi_l} = \mathbb{E}_{g_{(i+1)k} \sim \pi_h, s_{(i+1)k} \sim \tau_i^{\pi_h, \pi_l}} \left[ \mathcal{D}(\psi(s_{(i+1)k}), g_{(i+1)k}) / \mathcal{D}(\psi(s_{ik}), g_{(i+1)k}) \right]. \qquad (3.5)$$

Then, we derive that

$$L_2 \leq \frac{r_{max}}{(1-\gamma)^2} (\mathcal{R}_{max}^{\pi_h, \pi_l} + 2\gamma^k). \qquad (3.6)$$

Thus, we take the results of Equations (3.4) and (3.6) and achieve the final bound.

The variable $\mathcal{R}_i^{\pi_h, \pi_l}$ emerges as a pivotal element for several reasons. Firstly, it serves as a crucial bridge connecting high- and low-level policies. The denominator reflects subgoal guidance by $\pi_h$, while the numerator represents the final distance achieved through exploration by $\pi_l$. This dual perspective highlights its significance in the hierarchical framework. Secondly, in contrast to prior work where subgoal reachability relies on environmental dynamics (Zhang et al., 2020) or policy behavior (Nachum et al., 2018b; Kreidieh et al., 2019), our metric focuses on task completion, measured by the final distance divided by the initial distance. This shift of focus allows us to centre our attention squarely on the task itself. As a result, we propose $\mathcal{R}_i^{\pi_h, \pi_l}$ as a novel subgoal reachability metric in this paper, which lays the foundation for our algorithm design.

## 3.3 ALGORITHMIC INSTANTIATION

Motivated by theoretical insights in the previous section, here we turn to design the core mutual response mechanism, which prompts the hierarchical policy optimization. Instantiating mutual response mechanism amounts to specify two main designs: 1) how to measure the dynamically varying subgoal reachability; 2) how to incorporate subgoal reachability into the high-level policy optimization and low-level policy optimization?

**Efficient subgoal reachability computation.** Although equation (3.5) provides a method for computing subgoal reachability, the computational efficiency may be hindered when the low-level distance function $\mathcal{D}$ is complex. Fortunately, by recognizing that the low-level intrinsic reward shares the same form as the distance computation, we can replace the distance computation with the low-level reward. Thus, we can calculate the subgoal reachability by

$$\mathcal{R}_i^{\pi_h,\pi_l} = \mathbb{E}_{g_{(i+1)k}\sim\pi_h, s_{(i+1)k}\sim\tau_i^{\pi_h,\pi_l}} \left[ \frac{\mathcal{D}(\psi(s_{(i+1)k}), g_{(i+1)k})}{\mathcal{D}(\psi(s_{ik}), g_{(i+1)k})} \right] = \mathbb{E}_{r_l\sim\tau_i^{\pi_h,\pi_l}} \frac{r_{l,(i+1)k}}{r_{l,ik}}. \quad (3.7)$$

Specifically, we use a temporary replay buffer for storing subtask trajectory $\tau_i^{\pi_h,\pi_l}$ upon subtask completion. Then, we can sample the first low-level reward $r_{l,ik} = r_l(s_{ik}, a_{ik}, g_{(i+1)k})$ and the last one $r_{l,(i+1)k} = r_l(s_{(i+1)k}, a_{(i+1)k}, g_{(i+1)k})$ from the temporary buffer to calculate the reachability. If $r_{l,ik} = 0$, it indicates that the initial state has already reached the subgoal, and the subtask is deemed complete. Then, we set $\mathcal{R}_i^{\pi_h,\pi_l} = 0$, and $\pi_l$ can be inactive for the remainder of the subtask.

Notably, our design mitigates over-reliance on the summation of step-wise low-level reward or the environmental dynamic property, but merely considers the start and end point of the subtask trajectory, consequently relaxing the constraint on low-level policy exploration. Further, such a design is quite lightweight, incurring $O(1)$ computational complexity, without introducing additional training costs.

**High-level policy optimization.** In our approach, we opt to use $\mathcal{R}_i^{\pi_h,\pi_l}$ as a regularization for optimizing $\pi_h$ rather than reward penalties. This choice offers several advantages. During the high-level policy evaluation phase, we exclusively rely on rewards from the environment to iteratively compute Q-values, which ensures the accuracy of guidance performance evaluation. Furthermore, in the policy improvement phase, using $\mathcal{R}_i^{\pi_h,\pi_l}$ as the regularization explicitly constrains the high-level policy's subgoal generation. This focus allows it to concern the subgoal-reaching performance of the low-level policy within a subtask. Specifically, when using soft-actor-critic (SAC) (Haarnoja et al., 2018b) as the backbone, we evaluate the high-level policy in the usual manner,

$$Q^{\pi_h}(s,g) = \arg\min_Q \frac{1}{2} \mathbb{E}_{s,g\sim D_h} \left[ r_h(s,g) + \gamma\mathbb{E}_{s'\sim D_h, g'\sim\pi_h} Q^{\pi_h}(s',g') - Q^{\pi_h}(s,g) \right]^2, \quad (3.8)$$

while the policy updates by minimizing the expected KL-divergence with the reachability term as,

$$\pi_h = \arg\min_{\pi_h} \mathbb{E}_{s\sim D_h} \left[ \mathrm{D}_{KL}(\pi_h(\cdot|s) \| \exp(Q^{\pi_h}(s,g) - V^{\pi_h}(s))) + \lambda_1 \mathcal{R}_i^{\pi_h,\pi_l} \right], \quad (3.9)$$

where $V^{\pi_h}(s) = \mathbb{E}_{g\sim\pi_h(\cdot|s)} \left[ Q^{\pi_h}(s,a) - \log\pi_h(\cdot|s) \right]$ is the high-level soft state value function and $\lambda_1$ is a weight factor. Thus, we can adjust the response of the high level through tuning $\lambda_1$.

**Low-level policy optimization.** In contrast to high-level policy, we utilize $\mathcal{R}_i^{\pi_h,\pi_l}$ as a reward bonus for low-level policy. This approach is designed to enable $\pi_l$ to simultaneously focus on both low-level rewards and subgoal reachability during subgoal exploration. It's important to note that enhancing low-level rewards by $\pi_l$ can not guarantee improved subgoal reachability. For instance, in Subtask 1, a subgoal that is nearby yet challenging may result in poor subgoal reachability (e.g., initial distance is 10, but final distance is 5). Conversely, in Subtask 2, distant yet accessible subgoals can achieve better subgoal reachability (e.g., initial distance is 20, and final distance is 8). To address this, we introduce subgoal reachability as well as the low-level reward in low-level policy optimization, aiming to enhance the overall performance of $\pi_l$. Thus, the surrogate low-level reward is

$$\hat{r}_l(s_{ik+j}, a_{ik+j}, g_{(i+1)k}) = r_l(s_{ik+j}, a_{ik+j}, g_{(i+1)k}) - \lambda_2 \mathcal{R}_i^{\pi_h,\pi_l}. \quad (3.10)$$

With the surrogate low-level reward established, various policy optimizers (Fujimoto et al., 2018; Haarnoja et al., 2018b) could be adopted, here we also opt for SAC as the backbone algorithm for low-level policy optimization,

$$Q^{\pi_l}(s,a) = \arg\min_Q \frac{1}{2} \mathbb{E}_{s,g,a\sim D_l} \left[ \hat{r}_l(s,a,g) + \gamma\mathbb{E}_{s',g\sim D_l, a'\sim\pi_l} Q^{\pi_l}(s',a') - Q^{\pi_l}(s,a) \right]^2, \quad (3.11)$$

$$\pi_l = \arg\min_{\pi_l} \mathbb{E}_{s,g\sim D_l} \left[ \mathrm{D}_{KL}(\pi_l(\cdot|s,g) \| \exp(Q^{\pi_l}(s,a) - V^{\pi_l}(s))) \right]. \quad (3.12)$$

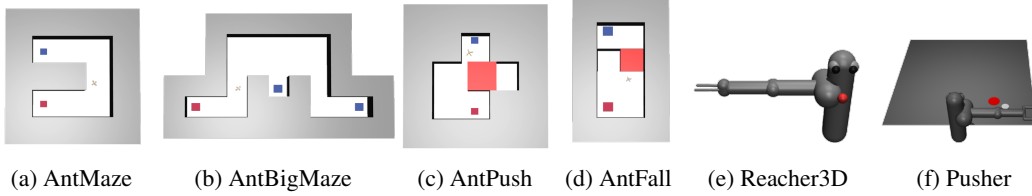

| (a) AntMaze | (b) AntBigMaze | (c) AntPush | (d) AntFall | (e) Reacher3D | (f) Pusher |

Figure 3: Environments used in our experiments. In maze tasks, the red square indicates the start point and the blue square represents the target point. In manipulation tasks, a robotic arm aims to make its end-effector and (puck-shaped) grey object reach the target position, which is marked as a red ball, respectively.

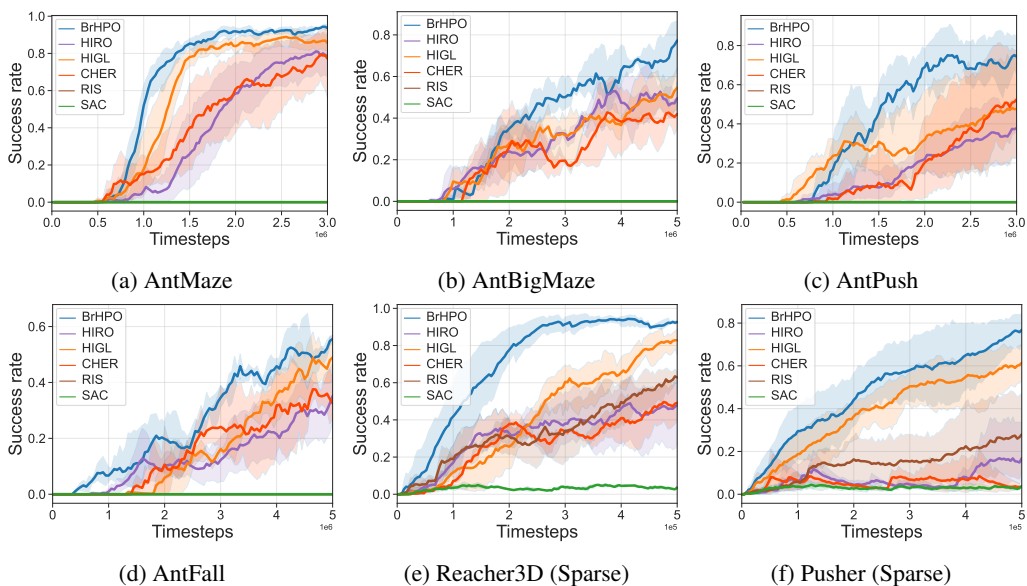

Figure 4: The average success rate in various continuous control tasks of BrHPO and baselines. The solid lines are the average success rate, while the shades indicate the standard error of the average performance. All algorithms are evaluated with 5 random seeds.

# 4 EXPERIMENT

Our experimental evaluation aims to investigate the following questions: 1) How does BrHPO's performance on long-term goal-conditioned benchmark tasks compare to that of state-of-the-art counterparts in terms of sample efficiency and asymptotic performance? 2) How effective is the mutual response mechanism in enhancing subgoal reachability and improving performance?

## 4.1 EXPERIMENTAL SETUP

We evaluate BrHPO on two categories of challenging long-horizon continuous control tasks, which feature both *dense* and *sparse* environmental reward, as illustrated in Figure 3. In the maze navigation environments, the reward is determined by the negative $\mathcal{L}_2$ distance between the current state and the target position within the goal space. In the robotics manipulation environments with sparse rewards, the reward is set to 0 when the distance is below a predefined threshold; otherwise, it's set to $-1$. Task success is defined as achieving a final distance to the target point of $d \leq 5$ for the maze tasks and $d \leq 0.25$ for the manipulation tasks. To ensure a fair comparison, all agents are initialized at the same position, eliminating extra environmental information introduction from random initialization, as discussed in (Lee et al., 2022). Detailed settings can be found in Appendix B.

## 4.2 COMPARATIVE EVALUATION

We compared BrHPO with the following baselines. 1) *HIRO* (Nachum et al., 2018b): designed an off-policy correction mechanism which required high-level experience to obey the current low-level policy; 2) *HIGL* (Kim et al., 2021): relied on the off-policy correction mechanism and introduced a $k$-step adjacent constraint (Zhang et al., 2020) and the novelty to discover appropriate subgoals; 3) *RIS* (Chane-Sane et al., 2021): utilized the hindsight method to generate the least-cost middle points as subgoals, forcing the low-level policy to follow the given subgoals; 4) *CHER* (Kreidieh et al., 2019): considered the cooperation of hierarchical policies, and the high-level policy needs to care about the low-level behaviour per step; 5) *SAC* (Haarnoja et al., 2018b): served as a benchmark of flat off-policy model-free algorithm and was applied as the backbone of BrHPO. Simply put, HIRO and HIGL focused on low-level domination, and RIS focused on high-level domination. CHER also considers the cooperation of different level policies while it requires step-by-step consideration.

The learning curves of BrHPO and the baselines across all tasks are plotted in Figure 4. Overall, the results demonstrate that BrHPO outperforms all baselines both in exploration efficiency and asymptotic performance. In particular, when dealing with large-scale (AntBigMaze) and partially-observed environments (AntPush and AntFall), BrHPO achieves better exploration and training stability, benefitting from the mutual response mechanism with information sharing and error correction for both levels. In contrast, acceptable baselines like HIRO, HIGL and CHER exhibit performance fluctuations and low success rates. It's worth noting that BrHPO can handle *sparse* reward environments without any reward shaping or hindsight relabeling modifications, indicating that our proposed mechanism can capture serendipitous success and provide intrinsic guidance.

## 4.3 ABLATION STUDY

Next, we make ablations and modifications to our method to validate the effectiveness and robustness of the mechanism we devised.

**Ablation on design choices.** To investigate the effectiveness of each component, we compared BrHPO with several variants through AntMaze and AntPush tasks. The BrHPO variants include, 1) *Vanilla*, which removes the mutual response mechanism in both-level policies, resulting in $\pi_h$ and $\pi_l$ being trained solely by conventional SAC; 2) *NoReg*, which keeps the low-level reward bonus but disables the regularization term in high-level policy training; 3) *NoBonus*, where only the high-level policy concerns subgoal reachability but the low-level reward bonus is removed.

The learning curves and state-subgoal trajectory visualizations from different variants are presented in Figure 6. BrHPO outperforms all three variants by a significant margin, highlighting the importance of the mutual response mechanism at both levels. Interestingly, the *NoBonus* variant achieves better performance than the *NoReg* variant, suggesting that the subgoal reachability addressed by the high-level policy has a greater impact on overall performance. This observation is further supported by the trajectory visualization results.

**Computation load.** The training wall-time is reported in Figure 5, where methods are benchmarked on a single RTX3090 GPU. Our method demonstrates efficient computational performance, with training times comparable to a flat SAC policy. Notably, compared to previous approaches that utilize adjacency matrices or graphs to model subgoal reachability, our method achieves at least a 2x improvement in training efficiency with performance guarantee. More details are provided in Table 4 of Appendix B.5.

**Hyperparameters.** We empirically studied the sensitivity of weight factors $\lambda_1$ and $\lambda_2$ in Figure 7. The results show that $\lambda_1$ and $\lambda_2$ within a certain range are acceptable. Upon closer analysis, we observed that when $\lambda_1$ is too small, the regularization term in high-level policy

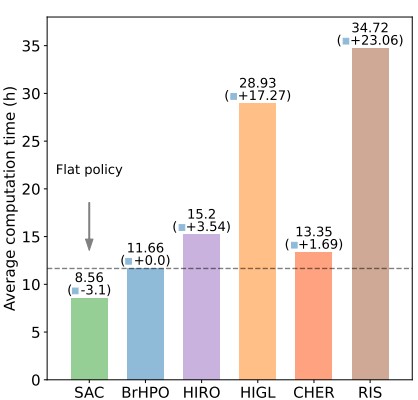

Figure 5: Training wall-time

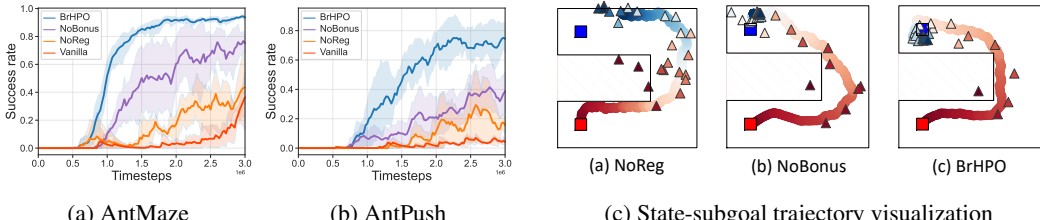

(a) AntMaze          (b) AntPush          (c) State-subgoal trajectory visualization

Figure 6: The performance and state-subgoal trajectory visualization from different BrHPO variants.

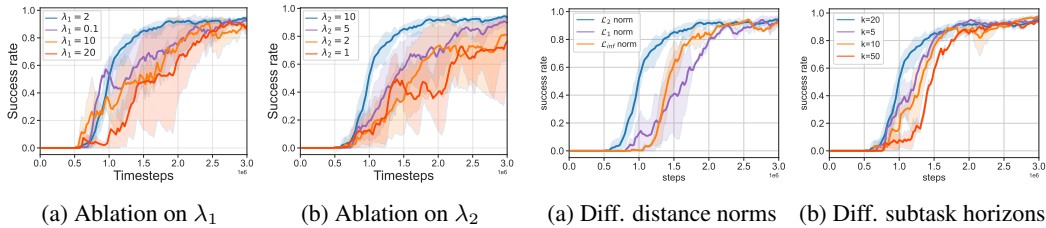

(a) Ablation on $\lambda_1$          (b) Ablation on $\lambda_2$          (a) Diff. distance norms          (b) Diff. subtask horizons

Figure 7: The learning curves with different weight factors $\lambda_1$ and $\lambda_2$ by AntMaze task.

Figure 8: The learning curves from different $\mathcal{D}$ and $k$ to verify the robustness of the mechanism.

optimization has minimal influence. Consequently, the high-level policy tends to disregard the performance of the low-level policy during tuning, resembling a high-level dominance scenario. Conversely, when $\lambda_1$ is too large, the high-level policy overly prioritizes subgoal reachability, diminishing its exploration capability and resembling a low-level dominance scenario. These observations validate the effectiveness of the mutual response mechanism in maintaining a balanced interaction between the high- and low-level policies. Additionally, the results for $\lambda_2$ suggest that a larger value can generally improve subgoal reachability from the perspective of the low-level policy, leading to performance improvements and enhanced stability.

**Robustness of mutual response mechanism.** We conducted additional experiments on the AntMaze task to verify the robustness of the proposed mechanism. The computation of subgoal reachability, a key factor in the mutual response mechanism, depends on the choice of the distance measurement $\mathcal{D}$ and the subtask horizon $k$. To test the distance measurement $\mathcal{D}$, we compared three distance functions: $\mathcal{L}_2$ norm, $\mathcal{L}_\infty$ norm, and $\mathcal{L}_1$ norm. Figure 8a shows that our method performs well regardless of the distance function used, highlighting the adaptability of the proposed mechanism. Additionally, we varied the subtask horizon by setting $k = 5, 10, 20, 50$ (Figure 8b). Surprisingly, we achieved success rates of around 0.9 with different subtask horizons, indicating that the performance is robust to variations in the subtask horizon, only with a slight effect on the convergence speed during training.

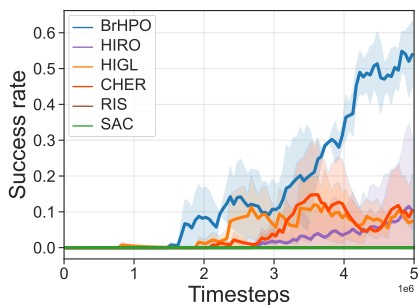

Figure 9: Performance comparison between BrHPO and baselines by HumanoidMaze. Mean and std by 4 runs.

This flexibility of BrHPO in decoupling the high- and low-level horizons without the need for extra graphs, as required in DHRL (Lee et al., 2022), is noteworthy. More ablations by Reacher3D task are provided in Figure 14 of Apppendix B.5.

In addition to evaluating parametric robustness, we subjected BrHPO to testing in stochastic environments to further evaluate its robustness. As depicted in Figure 15 of Apppendix B.5, we introduce varying levels of Gaussian noise into the state space. The results demonstrate our BrHPO can effectively mitigate the impact of noise and ensure consistent final performance.

**Extension in complex environment.** To further evaluate BrHPO's performance, we introduce a challenging HumanoidMaze task, where a humanoid robot navigates through a maze with right-angled turns. In this task, the simulated humanoid operates in a state space comprising 274 dimensions and

an action space of 17 dimensions. The primary goal of $\pi_l$ is to maintain body balance while following subgoals generated by the high-level policy. Consequently, extensive training is required for the $\pi_l$ to enable the humanoid to acquire proficient walking skills. This training process demands that $\pi_h$ exhibits "patience", gradually adjusting subgoals to effectively guide the humanoid's progress. As shown in Figure 9, the performance comparison clearly demonstrates the significant advantage of BrHPO over the baselines, even when dealing with high-dimensional continuous tasks. For additional insights, trajectory visualizations are available in Figure 12 of Appendix B.5.

## 5 RELATED WORKS

Hierarchical Reinforcement Learning (HRL) methods have emerged as promising solutions for addressing long-horizon complex tasks, primarily due to the synergistic collaboration between high-level task division and low-level exploration (Jong et al., 2008; Haarnoja et al., 2018a; Nachum et al., 2019; Pateria et al., 2021b; Eppe et al., 2022). Generally, HRL methods can be broadly categorized into two groups, option-based HRL (Sutton et al., 1999; Precup et al., 1998; Zhang et al., 2021; Mannor et al., 2004) and subgoal-based HRL (Dayan & Hinton, 1992; Nachum et al., 2019; Campos et al., 2020; Li et al., 2021b; Islam et al., 2022), that highlights the scope of guidance provided by the high-level policy. The first avenue in HRL involves the use of options to model the policy-switching mechanism in long-term tasks, which provides guidance to the low-level policy on when to terminate the current subtask and transition to a new one (Machado et al., 2017; Zhang & Whiteson, 2019). In contrast, the subgoal-based HRL avenue (Vezhnevets et al., 2017; Nachum et al., 2018a; Gürtler et al., 2021; Czechowski et al., 2021; Li et al., 2021a) focuses on generating subgoals in fixed horizon subtasks rather than terminal signals, and our work falls under this category. Notably, subgoal-based HRL approaches prioritize subgoal reachability as a means of achieving high performance (Stein et al., 2018; Paul et al., 2019; Li et al., 2020; Czechowski et al., 2021; Pateria et al., 2021a).

Various methods have been proposed to enhance subgoal reachability, from either the high-level or low-level perspectives. When the low level is considered to be dominant, several works have proposed relabeling or correcting subgoals based on the exploration capacity of the low-level policy. Examples include off-policy correction in HIRO (Nachum et al., 2018b) and hindsight relabeling in HER (Andrychowicz et al., 2017), RIS (Chane-Sane et al., 2021) and HAC (Levy et al., 2019). On the other hand, when the high-level dominates, subgoals are solved from given prior experience or knowledge, and the low-level policy is trained merely to track the given subgoals (Savinov et al., 2018; Huang et al., 2019; Eysenbach et al., 2019; Jurgenson et al., 2020). In contrast to the listed prior works, BrHPO proposes a mutual response mechanism for ensuring bidirectional reachability.

Meanwhile, our method relates to previous research that encourages cooperation between the high-level policy and the low-level one, where they explored various techniques for modelling subgoal reachability, including $k$-step adjacency matrix (Ferns et al., 2004; Castro, 2020; Zhang et al., 2020) or state-subgoal graph (Zhang et al., 2018; Kim et al., 2021; Lee et al., 2022). However, these methods can be computationally intensive and conservative. Our proposed method provides a more computationally efficient and flexible approach to gain subgoal reachability. By avoiding an explicit representation of the state-subgoal adjacency, our method can be more easily deployed and applied to a variety of different environments.

## 6 CONCLUSION

In this work, we identify that bilateral information sharing and error correction have been long neglected in previous HRL works. This will potentially cause local exploration and unattainable subgoal generation, which hinders overall performance and sample efficiency. To address this issue, we delve into the mutual response of hierarchical policies, both theoretically and empirically, revealing the crucial role of the mutual response mechanism. Based on these findings, we proposed the Bidirectional-reachable Hierarchical Policy Optimization (BrHPO) algorithm. BrHPO not only matches the best HRL algorithms in asymptotic performance, but it also shines in low computational load. Although BrHPO offers many advantages, a main challenge is to design an appropriate low-level reward to compute the subgoal reachability, thus limiting the application in sparse low-level reward settings (Lee et al., 2022). Future work that merits investigation are integrating up-to-date reachability measurement and policy optimization backbone to develop strong HRL algorithm.

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

# A THEORETICAL ANALYSIS

## A.1 OMITTED PROOFS

**Theorem A.1** (Sub-optimal performance difference bound of HRL). *The performance difference bound $C$ between the induced optimal hierarchical policies $\Pi^*$ and the learned one $\Pi$ can be*

$$C(\pi_h, \pi_l) = \frac{2r_{max}}{(1-\gamma)^2} \left[ \underbrace{(1+\gamma)\mathbb{E}_{g\sim\pi_h}\left(1+\frac{\pi_h^*}{\pi_h}\right)\epsilon_{\pi_l^*,\pi_l}^g}_{\text{(i) hierarchical policies' inconsistency}} + \underbrace{2\left(\mathcal{R}_{max}^{\pi_h,\pi_l} + 2\gamma^k\right)}_{\text{(ii) subgoal reachability penalty}} \right], \quad \text{(A.1)}$$

*where $\epsilon_{\pi_l^*,\pi_l}^g$ is the distribution shift between $\pi_l^*$ and $\pi_l$, and $\mathcal{R}_{max}^{\pi_h,\pi_l}$ is the maximum subgoal reachability penalty from the learned one $\Pi$, both of which are formulated as,*

$$\epsilon_{\pi_l^*,\pi_l}^g = \max_{s\in\mathcal{S},g\sim\pi_h} D_{TV}\left(\pi_l^*(\cdot|s,g)\|\pi_l(\cdot|s,g)\right) \quad \text{and} \quad \mathcal{R}_{max}^{\pi_h,\pi_l} = \max_{i\in\mathbb{N}} \mathcal{R}_i^{\pi_h,\pi_l}.$$

*Proof.* To derive the performance difference bound between $\Pi^*$ and $\Pi$, we first divide the bound into three terms,

$$V^{\Pi^*}(s_0) - V^{\Pi}(s_0) = V^{\pi_h^*,\pi_l^*}(s_0) - V^{\pi_h,\pi_l}(s_0)$$

$$= \underbrace{V^{\pi_h^*,\pi_l^*}(s_0) - V^{\pi_h^*,\pi_l}(s_0)}_{L_1} + \underbrace{V^{\pi_h^*,\pi_l}(s_0) - V^{\pi_h,\pi_l^*}(s_0)}_{L_2}$$

$$+ \underbrace{V^{\pi_h,\pi_l^*}(s_0) - V^{\pi_h,\pi_l}(s_0)}_{L_3}. \quad \text{(A.2)}$$

Then, our proof can be obtained to by tackling $L_1$, $L_2$ and $L_3$, respectively.

- **Derivation of $L_1$**

By adding and subtracting the same term in $L_1$, we obtain

$$L_1 = V^{\pi_h^*,\pi_l^*}(s_0) - \left[\widetilde{V}_0^{\pi_h^*,\pi_l^*}(s_0) + \gamma^k \mathbb{E}_{g\sim\pi_h^*,s\sim\mathbb{P}_k^{\pi_l^*,g}(\cdot|s_0)} V^{\pi_h^*,\pi_l}(s_k)\right]$$

$$+ \left[\widetilde{V}_0^{\pi_h^*,\pi_l^*}(s_0) + \gamma^k \mathbb{E}_{g\sim\pi_h^*,s\sim\mathbb{P}_k^{\pi_l^*,g}(\cdot|s_0)} V^{\pi_h^*,\pi_l}(s_k)\right] - V^{\pi_h^*,\pi_l}(s_0)$$

$$= \left[\widetilde{V}_0^{\pi_h^*,\pi_l^*}(s_0) + \gamma^k \mathbb{E}_{g\sim\pi_h^*,s\sim\mathbb{P}_k^{\pi_l^*,g}(\cdot|s_0)} V^{\pi_h^*,\pi_l^*}(s_k)\right] \quad \leftarrow \quad \text{\color{blue}By Lemma A.3}$$

$$- \left[\widetilde{V}_0^{\pi_h^*,\pi_l^*}(s_0) + \gamma^k \mathbb{E}_{g\sim\pi_h^*,s\sim\mathbb{P}_k^{\pi_l^*,g}(\cdot|s_0)} V^{\pi_h^*,\pi_l}(s_k)\right]$$

$$+ \left[\widetilde{V}_0^{\pi_h^*,\pi_l^*}(s_0) + \gamma^k \mathbb{E}_{g\sim\pi_h^*,s\sim\mathbb{P}_k^{\pi_l^*,g}(\cdot|s_0)} V^{\pi_h^*,\pi_l}(s_k)\right]$$

$$- \left[\widetilde{V}_0^{\pi_h^*,\pi_l}(s_0) + \gamma^k \mathbb{E}_{g\sim\pi_h^*,s\sim\mathbb{P}_k^{\pi_l,g}(\cdot|s_0)} V^{\pi_h^*,\pi_l}(s_k)\right]$$

$$= \underbrace{\gamma^k \mathbb{E}_{g\sim\pi_h^*,s\sim\mathbb{P}_k^{\pi_l^*,g}(\cdot|s_0)} \left[V^{\pi_h^*,\pi_l^*}(s_k) - V^{\pi_h^*,\pi_l}(s_k)\right]}_{\text{part a}} + \underbrace{\left[\widetilde{V}_0^{\pi_h^*,\pi_l^*}(s_0) - \widetilde{V}_0^{\pi_h^*,\pi_l}(s_0)\right]}_{\text{part b}}$$

$$+ \underbrace{\gamma^k \left[\mathbb{E}_{g\sim\pi_h^*,s\sim\mathbb{P}_k^{\pi_l^*,g}(\cdot|s_0)} V^{\pi_h^*,\pi_l}(s_k) - \mathbb{E}_{g\sim\pi_h^*,s\sim\mathbb{P}_k^{\pi_l,g}(\cdot|s_0)} V^{\pi_h^*,\pi_l}(s_k)\right]}_{\text{part c}}. \quad \text{(A.3)}$$

Then, we can deal with the three parts one by one to obtain the derivation of $L_1$. Note that, part b represents the performance discrepancy in the first subtask, caused by different low-level policies $\pi_l^*$ and $\pi_l$. Thus, consider the policy shift of the low-level policies, we suppose

$$\epsilon_{\pi_l^*,\pi_l}^g = \max_{s\in\mathcal{S},g\sim\pi_h} D_{TV}\left(\pi_l^*(\cdot|s,g)\|\pi_l(\cdot|s,g)\right). \quad \text{(A.4)}$$

Then, recall $r_{max}$ to be the maximum environmental reward, i.e., $r \leq r_{max}$, we have

$$\text{part b} = \widetilde{V}_0^{\pi_h^*, \pi_l^*}(s_0) - \widetilde{V}_0^{\pi_h^*, \pi_l}(s_0)$$

$$= \sum_{j=0}^{k-1} \mathbb{E}_{g_k \sim \pi_h^*, s, a \sim \mathbb{P}_j^{\pi_l^*, g}(\cdot, \cdot | s_0)} \left[ \gamma^j r(s_j, a_j, \hat{g}) \right] - \sum_{j=0}^{k-1} \mathbb{E}_{g_k \sim \pi_h^*, s, a \sim \mathbb{P}_j^{\pi_l, g}(\cdot, \cdot | s_0)} \left[ \gamma^j r(s_j, a_j, \hat{g}) \right]$$

$$\leq \sum_{j=0}^{k-1} \mathbb{E}_{g_k \sim \pi_h^*} 2 \left[ \gamma^j r(s_j, a_j, \hat{g}) \right] D_{TV} \left( \mathbb{P}_j^{\pi_l^*, g}(\cdot, \cdot | s_0) \middle\| \mathbb{P}_j^{\pi_l, g}(\cdot, \cdot | s_0) \right)$$

$$\leq 2 r_{max} \sum_{j=0}^{k-1} \mathbb{E}_{g_k \sim \pi_h^*} \gamma^j j \epsilon_{\pi_l^*, \pi_l}^g. \qquad \leftarrow \quad \text{By Lemma A.4} \tag{A.5}$$

For part c, note that the joint value function can be bounded as $V^{\pi_h, \pi_l}(s_0) \leq r_{max}/(1 - \gamma)$. We can apply Lemma A.4 to bound the discrepancy of the low-level policies, and have

$$\text{part c} = \gamma^k \left[ \mathbb{E}_{g \sim \pi_h^*, s \sim \mathbb{P}_k^{\pi_l^*, g}(\cdot | s_0)} V^{\pi_h^*, \pi_l}(s_k) - \mathbb{E}_{g \sim \pi_h^*, s \sim \mathbb{P}_k^{\pi_l, g}(\cdot | s_0)} V^{\pi_h^*, \pi_l}(s_k) \right]$$

$$= \gamma^k \int_{g \in \mathcal{G}} \int_{s \in \mathcal{S}} \pi_h^*(g | s_k) \left( \mathbb{P}_k^{\pi_l^*, g}(s | s_0) - \mathbb{P}_k^{\pi_l, g}(s | s_0) \right) V^{\pi_h^*, \pi_l}(s) \, ds \, dg$$

$$\leq \frac{2\gamma^k r_{max}}{1 - \gamma} \mathbb{E}_{g \sim \pi_h^*} \left[ D_{TV} \left( \mathbb{P}_k^{\pi_l^*, g}(\cdot | s_0) \| \mathbb{P}_k^{\pi_l, g}(\cdot | s_0) \right) \right]$$

$$\leq \frac{2\gamma^k r_{max}}{1 - \gamma} \mathbb{E}_{g \sim \pi_h^*} k \epsilon_{\pi_l^*, \pi_l}^g, \tag{A.6}$$

At last, for part a, we can apply the same recursion every $k$ steps,

$$\text{part a} = \gamma^k \mathbb{E}_{g \sim \pi_h^*, s \sim \mathbb{P}_k^{\pi_l^*, g}(\cdot | s_0)} \left[ V^{\pi_h^*, \pi_l^*}(s_k) - V^{\pi_h^*, \pi_l}(s_k) \right]$$

$$\leq \gamma^{2k} \mathbb{E}_{g \sim \pi_h^*, s \sim \mathbb{P}_{2k}^{\pi_l^*, g}(\cdot | s_0)} \left[ V^{\pi_h^*, \pi_l^*}(s_{2k}) - V^{\pi_h^*, \pi_l}(s_{2k}) \right]$$

$$+ 2 r_{max} \sum_{j=k}^{2k-1} \mathbb{E}_{g \sim \pi_h^*} \gamma^j j \epsilon_{\pi_l^*, \pi_l}^g + \frac{2\gamma^{2k} r_{max}}{1 - \gamma} \mathbb{E}_{g \sim \pi_h^*} 2k \epsilon_{\pi_l^*, \pi_l}^g. \tag{A.7}$$

Now, with the derivation of part a, part b and part c, we can combine these and repeat the recursion step for infinitely many times

$$L_1 = \text{part a} + \text{part b} + \text{part c}$$

$$\leq 2 r_{max} \sum_{j=0}^{k-1} \mathbb{E}_{g_k \sim \pi_h^*} \gamma^j j \epsilon_{\pi_l^*, \pi_l}^g + \frac{2\gamma^k r_{max}}{1 - \gamma} \mathbb{E}_{g \sim \pi_h^*} k \epsilon_{\pi_l^*, \pi_l}^g$$

$$+ 2 r_{max} \sum_{j=k}^{2k-1} \mathbb{E}_{g_{2k} \sim \pi_h^*} \gamma^j j \epsilon_{\pi_l^*, \pi_l}^g + \frac{2\gamma^{2k} r_{max}}{1 - \gamma} \mathbb{E}_{g \sim \pi_h^*} 2k \epsilon_{\pi_l^*, \pi_l}^g$$

$$+ \gamma^{2k} \mathbb{E}_{g \sim \pi_h^*, s \sim \mathbb{P}_{2k}^{\pi_l^*, g}(\cdot | s_0)} \left[ V^{\pi_h^*, \pi_l^*}(s_{2k}) - V^{\pi_h^*, \pi_l}(s_{2k}) \right]$$

$$\vdots$$

$$\leq 2 r_{max} \sum_{i=0}^{\infty} \sum_{j=0}^{k-1} \mathbb{E}_{g \sim \pi_h^*} \gamma^{(ik+j)} (ik + j) \epsilon_{\pi_l^*, \pi_l}^g + \frac{\gamma^{(i+1)k}}{1 - \gamma} \mathbb{E}_{g \sim \pi_h^*} (i + 1) k \epsilon_{\pi_l^*, \pi_l}^g$$

$$\leq 2 r_{max} \frac{1 + \gamma}{(1 - \gamma)^2} \mathbb{E}_{g \sim \pi_h^*} \epsilon_{\pi_l^*, \pi_l}^g. \tag{A.8}$$

Thus, we complete the derivation of $L_1$.

- **Derivation of $L_3$**

Compared with $L_1$, the term $L_3$ replaces the high-level policy from $\pi_h^*$ to $\pi_h$. Thus, we directly can get $L_3$ from the results of $L_1$ as

$$L_3 \leq 2r_{max}\frac{1+\gamma}{(1-\gamma)^2}\mathbb{E}_{g\sim\pi_h}\epsilon_{\pi_l^*,\pi_l}^g. \tag{A.9}$$

- **Derivation of $L_2$**

Similar to the derivation of $L_1$, by adding and subtracting the same term in $L_2$, we have

$$
\begin{aligned}
L_2 &= V^{\pi_h^*,\pi_l}(s_0) - \left[\widetilde{V}_0^{\pi_h^*,\pi_l}(s_0) + \gamma^k\mathbb{E}_{g\sim\pi_h^*,s\sim\mathbb{P}_k^{\pi_l,g}(\cdot|s_0)}V^{\pi_h,\pi_l^*}(s_k)\right] \\
&\quad + \left[\widetilde{V}_0^{\pi_h^*,\pi_l}(s_0) + \gamma^k\mathbb{E}_{g\sim\pi_h^*,s\sim\mathbb{P}_k^{\pi_l,g}(\cdot|s_0)}V^{\pi_h,\pi_l^*}(s_k)\right] - V^{\pi_h,\pi_l^*}(s_0) \\
&= \left[\widetilde{V}_0^{\pi_h^*,\pi_l}(s_0) + \gamma^k\mathbb{E}_{g\sim\pi_h^*,s\sim\mathbb{P}_k^{\pi_l,g}(\cdot|s_0)}V^{\pi_h^*,\pi_l}(s_k)\right] \\
&\quad - \left[\widetilde{V}_0^{\pi_h^*,\pi_l}(s_0) + \gamma^k\mathbb{E}_{g\sim\pi_h^*,s\sim\mathbb{P}_k^{\pi_l,g}(\cdot|s_0)}V^{\pi_h,\pi_l^*}(s_k)\right] \\
&\quad\quad + \left[\widetilde{V}_0^{\pi_h^*,\pi_l}(s_0) + \gamma^k\mathbb{E}_{g\sim\pi_h^*,s\sim\mathbb{P}_k^{\pi_l,g}(\cdot|s_0)}V^{\pi_h,\pi_l^*}(s_k)\right] \\
&\quad\quad\quad - \left[\widetilde{V}_0^{\pi_h,\pi_l^*}(s_0) + \gamma^k\mathbb{E}_{g\sim\pi_h,s\sim\mathbb{P}_k^{\pi_l^*,g}(\cdot|s_0)}V^{\pi_h,\pi_l^*}(s_k)\right] \\
&= \underbrace{\gamma^k\mathbb{E}_{g\sim\pi_h^*,s\sim\mathbb{P}_k^{\pi_l,g}(\cdot|s_0)}\left[V^{\pi_h^*,\pi_l}(s_k) - V^{\pi_h,\pi_l^*}(s_k)\right]}_{\text{part d}} + \underbrace{\left[\widetilde{V}_0^{\pi_h^*,\pi_l}(s_0) - \widetilde{V}_0^{\pi_h,\pi_l^*}(s_0)\right]}_{\text{part e}} \\
&\quad + \underbrace{\gamma^k\left[\mathbb{E}_{g\sim\pi_h^*,s\sim\mathbb{P}_k^{\pi_l,g}(\cdot|s_0)}V^{\pi_h,\pi_l^*}(s_k) - \mathbb{E}_{g\sim\pi_h,s\sim\mathbb{P}_k^{\pi_l^*,g}(\cdot|s_0)}V^{\pi_h,\pi_l^*}(s_k)\right]}_{\text{part f}}. \tag{A.10}
\end{aligned}
$$

According to Assumption A.5, we suppose $r(s_t, a_t, \hat{g}) = \mathbb{E}_{g\sim\pi_h,s,a\sim\mathbb{P}_t^{\pi_l,g}}r_l(s_t, a_t, g)/\mathcal{D}(g, \hat{g})$, thus we summate the $k$-step reward in the first subtask in part e as

$$
\begin{aligned}
&\sum_{j=0}^{k-1}\mathbb{E}_{g\sim\pi_h,s,a\sim\mathbb{P}_j^{\pi_l,g}(\cdot,\cdot|s_0)}\left[\gamma^j r(s_j, a_j, \hat{g})\right] \\
&= r(s_0, a_0, \hat{g})\sum_{j=0}^{k-1}\mathbb{E}_{g\sim\pi_h,s,a\sim\mathbb{P}_j^{\pi_l,g}(\cdot,\cdot|s_0)}\left[\gamma^j\frac{r(s_j, a_j, \hat{g})}{r(s_0, a_0, \hat{g})}\right] \\
&= r(s_0, a_0, \hat{g})\sum_{j=0}^{k-1}\mathbb{E}_{g\sim\pi_h,s,a\sim\mathbb{P}_j^{\pi_l,g}(\cdot,\cdot|s_0)}\left[\gamma^j\frac{r_l(s_j, a_j, g)}{\mathcal{D}(g, \hat{g})}\frac{\mathcal{D}(g, \hat{g})}{r_l(s_0, a_0, g)}\right] \\
&= r(s_0, a_0, \hat{g})\sum_{j=0}^{k-1}\mathbb{E}_{g\sim\pi_h,s,a\sim\mathbb{P}_j^{\pi_l,g}(\cdot,\cdot|s_0)}\left[\gamma^j\frac{r_l(s_j, a_j, g)}{r_l(s_0, a_0, g)}\right]. \tag{A.11}
\end{aligned}
$$

Since the low-level policy is trained as a goal-conditioned policy, we have $r_l(s_j, a_j, g) \leq r_l(s_k, a_k, g)$. And the summation in the first subtask can be

$$
\begin{aligned}
&\sum_{j=0}^{k-1}\mathbb{E}_{g\sim\pi_h,s,a\sim\mathbb{P}_j^{\pi_l,g}(\cdot,\cdot|s_0)}\left[\gamma^j r(s_j, a_j, \hat{g})\right] \\
&\leq r(s_0, a_0, \hat{g})\sum_{j=0}^{k-1}\mathbb{E}_{g\sim\pi_h,s,a\sim\mathbb{P}_j^{\pi_l,g}(\cdot,\cdot|s_0)}\left[\gamma^j\frac{r_l(s_k, a_k, g)}{r_l(s_0, a_0, g)}\right] \\
&= r(s_0, a_0, \hat{g})\frac{1-\gamma^k}{1-\gamma}\frac{r_l(s_k, a_k, g)}{r_l(s_0, a_0, g)}. \tag{A.12}
\end{aligned}
$$

Thus, we let the fraction $\mathcal{R}_i^{\pi_h,\pi_l} = r_l(s_k,a_k,g)/r_l(s_0,a_0,g)$ be the subgoal reachability definition, and the part e in $L_2$ can be

$$
\begin{aligned}
\text{part e} &= \widetilde{V}_0^{\pi_h^*,\pi_l}(s_0) - \widetilde{V}_0^{\pi_h,\pi_l^*}(s_0) \\
&= \sum_{j=0}^{k-1} \mathbb{E}_{g\sim\pi_h^*,s,a\sim\mathbb{P}_j^{\pi_l,g}(\cdot,\cdot|s_0)}\left[\gamma^j r(s_j,a_j,\hat{g})\right] - \sum_{j=0}^{k-1}\mathbb{E}_{g\sim\pi_h,s,a\sim\mathbb{P}_j^{\pi_l^*,g}(\cdot,\cdot|s_0)}\left[\gamma^j r(s_j,a_j,\hat{g})\right] \\
&\leq r(s_0,a_0,\hat{g})\frac{1-\gamma^k}{1-\gamma}\mathcal{R}_0^{\pi_h^*,\pi_l} - \sum_{j=0}^{k-1}\mathbb{E}_{g\sim\pi_h,s,a\sim\mathbb{P}_j^{\pi_l^*,g}(\cdot,\cdot|s_0)}\left[\gamma^j r(s_j,a_j,\hat{g})\right] \\
&\leq r_{max}\frac{1-\gamma^k}{1-\gamma}\left(\mathcal{R}_0^{\pi_h,\pi_l} - \mathcal{R}_0^{\pi_h^*,\pi_l^*}\right) \qquad \leftarrow \quad \Pi^* \text{ can achieve best subgoal reachability} \\
&\leq r_{max}\frac{1-\gamma^k}{1-\gamma}\mathcal{R}_0^{\pi_h,\pi_l}.
\end{aligned}
\tag{A.13}
$$

The penultimate inequality is based on the property of the induced optimal hierarchical policies. Compared with the learned $\pi_h$, Figure 3 shows that $\pi_h^*$ can balance the subgoal reachability and the guidance, thus $\mathcal{R}_0^{\pi_h,\pi_l} \geq \mathcal{R}_0^{\pi_h^*,\pi_l}$ (note that the smaller $\mathcal{R}$ implies the better subgoal reachability). And, the optimal policies $\Pi^*$ can achieve the optimal subgoal reachability, i.e. $\mathcal{R}_0^{\pi_h^*,\pi_l^*} \leq \mathcal{R}_0^{\pi_h,\pi_l}$. Thus, we have $\left(\mathcal{R}_0^{\pi_h^*,\pi_l} - \mathcal{R}_0^{\pi_h,\pi_l^*}\right) \leq \left(\mathcal{R}_0^{\pi_h,\pi_l} - \mathcal{R}_0^{\pi_h^*,\pi_l^*}\right)$.

Then, we turn to part f in $L_2$. Consider the upper bound of joint value function, we have

$$
\begin{aligned}
\text{part f} &= \gamma^k\left[\mathbb{E}_{g\sim\pi_h^*,s\sim\mathbb{P}_k^{\pi_l,g}(\cdot|s_0)}V^{\pi_h,\pi_l^*}(s_k) - \mathbb{E}_{g\sim\pi_h,s\sim\mathbb{P}_k^{\pi_l^*,g}(\cdot|s_0)}V^{\pi_h,\pi_l^*}(s_k)\right] \\
&\leq \gamma^k\int_{g\in\mathcal{G}}\int_{s\in\mathcal{S}}\left[\pi_h^*(g|s) - \pi_h(g|s)\right]\left[\mathbb{P}_k^{\pi_l,g}(s|s_0) - \mathbb{P}_k^{\pi_l^*,g}(s|s_0)\right]\frac{r_{max}}{1-\gamma}\mathrm{d}s\mathrm{d}g \\
&\leq 2\gamma^k\int_{g\in\mathcal{G}}\int_{s\in\mathcal{S}}\frac{r_{max}}{1-\gamma}\mathrm{d}s\mathrm{d}g \\
&= \frac{2\gamma^k r_{max}}{1-\gamma}.
\end{aligned}
\tag{A.14}
$$

With the derivation of part e and part f, we deal with part d by the recursion each $k$-steps as

$$
\begin{aligned}
\text{part d} &= \gamma^k\mathbb{E}_{g\sim\pi_h^*,s\sim\mathbb{P}_k^{\pi_l,g}(\cdot|s_0)}\left[V^{\pi_h^*,\pi_l}(s_k) - V^{\pi_h,\pi_l^*}(s_k)\right] \\
&\leq \gamma^{2k}\mathbb{E}_{g\sim\pi_h^*,s\sim\mathbb{P}_{2k}^{\pi_l,g}(\cdot|s_0)}\left[V^{\pi_h^*,\pi_l}(s_{2k}) - V^{\pi_h,\pi_l^*}(s_{2k})\right] \\
&\quad + r_{max}\frac{\gamma^k-\gamma^{2k}}{1-\gamma}\mathcal{R}_1^{\pi_h,\pi_l} + \frac{2\gamma^{2k}r_{max}}{1-\gamma}.
\end{aligned}
\tag{A.15}
$$

Thus, we combine the result of part d, part e and part f to obtain the results of $L_2$ as

$$
\begin{aligned}
L_2 &= \text{part d} + \text{part e} + \text{part f} \\
&\leq r_{max}\frac{1-\gamma^k}{1-\gamma}\mathcal{R}_0^{\pi_h,\pi_l} + r_{max}\frac{\gamma^k-\gamma^{2k}}{1-\gamma}\mathcal{R}_1^{\pi_h,\pi_l} + \frac{2\gamma^k r_{max}}{1-\gamma} + \frac{2\gamma^{2k}r_{max}}{1-\gamma} \\
&\quad + \gamma^{2k}\mathbb{E}_{g\sim\pi_h^*,s\sim\mathbb{P}_{2k}^{\pi_l,g}(\cdot|s_0)}\left[V^{\pi_h^*,\pi_l}(s_{2k}) - V^{\pi_h,\pi_l^*}(s_{2k})\right] \\
&\;\;\vdots \\
&\leq r_{max}\sum_{i=0}^{\infty}\frac{(1-\gamma^k)\gamma^{ik}}{1-\gamma}\mathcal{R}_i^{\pi_h,\pi_l} + \frac{2\gamma^{(i+1)k}}{1-\gamma} \\
&\leq \frac{r_{max}}{(1-\gamma)^2}\left(\mathcal{R}_{max}^{\pi_h,\pi_l} + 2\gamma^k\right).
\end{aligned}
\tag{A.16}
$$

In the last inequality, we define

$$\mathcal{R}_{max}^{\pi_h,\pi_l} = \max_{i\in\mathbb{N}} \mathcal{R}_i^{\pi_h,\pi_l}. \tag{A.17}$$

Now, we have the results of $L_1$, $L_2$ and $L_3$. The performance difference bound between $\Pi^*$ and $\Pi$ can be obtained as

$$
\begin{aligned}
V^{\Pi^*}(s_0) - V^{\Pi}(s_0) &= L_1 + L_2 + L_3 \\
&\leq 2r_{max}\frac{1+\gamma}{(1-\gamma)^2}\mathbb{E}_{g\sim\pi_h^*}\epsilon_{\pi_l^*,\pi_l}^g + \frac{r_{max}}{(1-\gamma)^2}(\mathcal{R}_{max}^{\pi_h,\pi_l} + 2\gamma^k) \\
&\quad + 2r_{max}\frac{1+\gamma}{(1-\gamma)^2}\mathbb{E}_{g\sim\pi_h}\epsilon_{\pi_l^*,\pi_l}^g \\
&= \frac{2r_{max}}{(1-\gamma)^2}\left[(1+\gamma)\mathbb{E}_{g\sim\pi_h}\left(1+\frac{\pi_h^*}{\pi_h}\right)\epsilon_{\pi_l^*,\pi_l}^g + 2\left(\mathcal{R}_{max}^{\pi_h,\pi_l} + 2\gamma^k\right)\right]. \tag{A.18}
\end{aligned}
$$

And the proof is complete. □

**Proposition A.2** (Equivalence between $\pi^*$ and $\Pi^*$). *With the $k$-step trajectory slicing and the alignment method, the performance of $\Pi^*$ and $\pi^*$ is equivalent, i.e., $V^{\pi^*}(s) = V^{\Pi^*}(s)$.*

*Proof.* According to the $k$-step trajectory slicing and the alignment method, the induced optimal hierarchical policies $\Pi^*$ can be generated by aligning with the $k$-step trajectory slice derived by $\pi^*$, thus we have

$$
\begin{aligned}
g_{(i+1)k} \sim \pi_h^*(\cdot|s_{ik}) &= \mathbb{P}_k^{\pi^*}(s_{(i+1)k}|s_{ik}) \\
&= p(s_{ik})\prod_{j=0}^{k-1}P(s_{ik+j+1}|s_{ik+j},a_{ik+j})\pi^*(a_{ik+j}|s_{ik+j}), \tag{A.19}
\end{aligned}
$$

$$a_{ik+j} \sim \pi_l^*(\cdot|s_{ik+j},g_{(i+1)k}) = \pi^*(a_{ik+j}|s_{ik+j}). \tag{A.20}$$

Thus, the value function for $\pi^*$ and the joint value function for $\Pi^*$ can be

$$
\begin{aligned}
V^{\pi^*}(s_0) &= \sum_t^\infty \gamma^t\mathbb{E}_{s\sim p(s'|s,a),a\sim\pi^*}\left[r(s_t,a_t,\hat{g})\right] \\
&= \sum_{i=0}^\infty\sum_{j=0}^{k-1}\mathbb{E}_{s\sim p(s'|s,a),a\sim\pi^*}\gamma^{ik+j}\left[r(s_{ik+j},a_{ik+j},\hat{g})\right] \\
&= \sum_{i=0}^\infty\mathbb{E}_{g\sim\pi_h^*}\left\{\gamma^{ik}\sum_{j=0}^{k-1}\mathbb{E}_{s\sim p(s'|s,a),a\sim\pi_l^*}\gamma^j\left[r(s_{ik+j},a_{ik+j},\hat{g})\right]\right\} \\
&= \sum_{i=0}^\infty\mathbb{E}_{g\sim\pi_h^*(\cdot|s)}\left[\gamma^{ik}\left(\sum_{j=0}^{k-1}\gamma^j\mathbb{E}_{s,a\sim\mathbb{P}_{ik+j}^{\pi_l^*,g}(\cdot,\cdot|s_0)}r(s_{ik+j},a_{ik+j},\hat{g})\right)\right] \\
&= V^{\Pi^*}(s_0) \tag{A.21}
\end{aligned}
$$

Thus, through the $k$-step trajectory slicing and the alignment method, the performance of $\Pi^*$ and $\pi^*$ is equivalent. And the proof is complete. □

## A.2 USEFUL LEMMA AND ASSUMPTION

**Lemma A.3** (Bellman Backup in HRL). *Consider that the joint value function can be decomposed by the summation of subtasks. Given the initial state $s_{ik}$ at the $i$-th subtask, the Bellman Backup of HRL defined in each subtask can be*

$$V^{\pi_h,\pi_l}(s_{ik}) = \widetilde{V}_i^{\pi_h,\pi_l}(s_{ik}) + \gamma^k\mathbb{E}_{g\sim\pi_h,s\sim\mathbb{P}_{(i+1)k}^{\pi_l,g}(\cdot|s_{ik})}\left[V^{\pi_h,\pi_l}(s_{(i+1)k})\right], \tag{A.22}$$

where $\widetilde{V}_i^{\pi_h,\pi_l}(s_{ik})$ is the the environment return of $\Pi$ with the $i$-th subtask, formulated as

$$\widetilde{V}_i^{\pi_h,\pi_l}(s_{ik}) = \sum_{j=0}^{k-1} \mathbb{E}_{g\sim\pi_h,s,a\sim\mathbb{P}_{ik+j}^{\pi_l,g}(\cdot,\cdot|s_{ik})} \left[ \gamma^j r(s_{ik+j}, a_{ik+j}, \hat{g}) \right]. \tag{A.23}$$

*Proof.* According to the decomposition of the joint value function $V^{\pi_h,\pi_l}(s)$, we have

$$V^{\pi_h,\pi_l}(s_0) = \sum_{i=0}^{\infty} \mathbb{E}_{g\sim\pi_h} \left[ \gamma^{ik} \left( \sum_{j=0}^{k-1} \gamma^j \mathbb{E}_{s,a\sim\mathbb{P}_{ik+j}^{\pi_l,g}(\cdot,\cdot|s_0)} r(s_{ik+j}, a_{ik+j}, \hat{g}) \right) \right]$$

$$= \sum_{j=0}^{k-1} \mathbb{E}_{g\sim\pi_h,s,a\sim\mathbb{P}_j^{\pi_l,g}(\cdot,\cdot|s_0)} \left[ \gamma^j r(s_j, a_j, \hat{g}) \right]$$

$$+ \sum_{i=1}^{\infty} \mathbb{E}_{g\sim\pi_h} \left[ \gamma^{ik} \left( \sum_{j=0}^{k-1} \gamma^j \mathbb{E}_{s,a\sim\mathbb{P}_{ik+j}^{\pi_l,g}(\cdot,\cdot|s_0)} r(s_{ik+j}, a_{ik+j}, \hat{g}) \right) \right]$$

$$= \widetilde{V}_0^{\pi_h,\pi_l}(s_0) + \gamma^k \mathbb{E}_{g\sim\pi_h,s\sim\mathbb{P}_k^{\pi_l,g}(\cdot|s_k)} \left[ V^{\pi_h,\pi_l}(s_k) \right]. \tag{A.24}$$

Thus, we can conclude that

$$V^{\pi_h,\pi_l}(s_{ik}) = \widetilde{V}_i^{\pi_h,\pi_l}(s_{ik}) + \gamma^k \mathbb{E}_{g\sim\pi_h,s\sim\mathbb{P}_{(i+1)k}^{\pi_l,g}(\cdot|s_{ik})} \left[ V^{\pi_h,\pi_l}(s_{(i+1)}) \right]. \tag{A.25}$$

And the proof is complete. $\square$

**Lemma A.4** (Markov chain TVD bound, time-varying). *Suppose the expected KL-divergence between two policy distributions is bounded as $\epsilon_{\pi_l^*,\pi_l}^g = \max_{s\in\mathcal{S},g\sim\pi_h} \mathrm{D}_{TV}\left(\pi_l^*(\cdot|s,g)\|\pi_l(\cdot|s,g)\right)$, and the initial state distributions are the same. Then, the distance in the state-action marginal is bounded as,*

$$\mathrm{D}_{TV}\left(\mathbb{P}_t^{\pi_l^*,g}(\cdot,\cdot|s_0)\Big\|\mathbb{P}_t^{\pi_l,g}(\cdot,\cdot|s_0)\right) \leq t\epsilon_{\pi_l^*,\pi_l}^g \tag{A.26}$$

*Proof.* Let $p(s'|s)$ as the adjacent state transition probability, which can be defined as

$$p(s'|s) = p(s)P(s'|s,a)\pi(a|s). \tag{A.27}$$

Replacing the policy as the low-level policy $\pi_l$, we can derive the Markov chain TVD bound caused by the different low-level policy,

$$\max_t \mathbb{E}_{s\sim p_1^t(s)} \mathrm{D}_{KL}(p_1(s'|s)\|p_2(s'|s))$$

$$= \max_t \mathbb{E}_{s\sim p_1^t(s)} p(s)P_{s,s'}^a(s'|s,a)\mathrm{D}_{KL}(\pi_l^*(a|s,g)\|\pi_l(a|s,g))$$

$$\leq \max_t \mathbb{E}_{s\sim p_1^t(s)} \mathrm{D}_{KL}(\pi_l^*(a|s,g)\|\pi_l(a|s,g))$$

$$\leq \max_{s\in\mathcal{S},g\sim\pi_h} \mathrm{D}_{TV}\left(\pi_l^*(\cdot|s,g)\|\pi_l(\cdot|s,g)\right)$$

$$= \epsilon_{\pi_l^*,\pi_l}^g \tag{A.28}$$

Thus, follow the Lemma B.2 in Janner et al. (2019), the distance in the state-action marginal is bounded as,

$$\mathrm{D}_{TV}\left(\mathbb{P}_t^{\pi_l^*,g}(\cdot,\cdot|s_0)\Big\|\mathbb{P}_t^{\pi_l,g}(\cdot,\cdot|s_0)\right) \leq t\epsilon_{\pi_l^*,\pi_l}^g. \tag{A.29}$$

And the proof is complete. $\square$

**Assumption A.5** (Refer to Assumption 1 in Zhang et al. (2022)). *For all $s\in\mathcal{S}$ and $g\in\mathcal{G}$, the environmental reward can be written as*

$$r(s,a,\hat{g}) = \sum_{s'} P_{s,s'}^a(s'|s,a)\pi_l(a|s,g)\widetilde{r}(s,s') = \mathbb{E}_{g\sim\pi_h,s,a\sim\mathbb{P}_t^{\pi_l,g}} r_l(s_t,a_t,g)/\mathcal{D}(g,\hat{g}). \tag{A.30}$$

*where $\widetilde{r}: \mathcal{S}\times\mathcal{G} \to [0, r_{max}]$ is a state-reachability reward function.*

In this assumption, the subgoal $g$ generated by the high-level policy represents the desired state to be reached, while the intermediate low-level state and action details are controlled by the low-level policy. Therefore, consider that the subgoals are generated towards the environmental goal $\hat{g}$, when given a low-level optimal/learned policy, it is natural to assume that the $k$-step stage reward only depends on the state where the agent starts and the state where the agent arrives.

# B   EXPERIMENTAL DETAILS

## B.1   IMPLEMENTATION DETAILS

Our method BrHPO and all baselines are implemented based on PyTorch.

**BrHPO.**   We employ the soft actor-critic (SAC) algorithm Haarnoja et al. (2018b) as the backbone framework for both high- and low-level policies. For the high-level policy, considering that the subtask trajectory $\tau_i^{\pi_h, \pi_l}$ in each subtask would be abstracted as one transition in high level, we convert the trajectory $(s_{ik:(i+1)k-1}, a_{ik:(i+1)k-1}, g_{(i+1)k}, r_{h,ik}, s_{(i+1)k})$ into a high-level transition tuple $(s_{ik}, g_{(i+1)k}, r_{h,ik}, s_{(i+1)k})$. Then, when a subtask ends, we compute the subgoal reachability by

$$\mathcal{R}_i^{\pi_h, \pi_l} = \mathbb{E}_{r_l \sim \tau_i^{\pi_h, \pi_l}} \frac{r_{l,(i+1)k}}{r_{l,ik}}.$$

Then, we can optimize the high-level policy by

$$Q^{\pi_h}(s, g) = \arg\min_Q \frac{1}{2} \mathbb{E}_{s,g \sim D_h} \left[ r_h(s, g) + \gamma \mathbb{E}_{s' \sim D_h, g' \sim \pi_h} Q^{\pi_h}(s', g') - Q^{\pi_h}(s, g) \right]^2,$$

$$\pi_h = \arg\min_{\pi_h} \mathbb{E}_{s \sim D_h} \left[ D_{KL}(\pi_h(\cdot|s) \| \exp(Q^{\pi_h}(s, g) - V^{\pi_h}(s))) + \lambda_1 \mathcal{R}_i^{\pi_h, \pi_l} \right].$$

For the low-level policy which can be trained as a goal-conditioned one, we design the reachability-aware low-level policy as

$$\hat{r}_l(s_{ik+j}, a_{ik+j}, g_{(i+1)k}) = r_l(s_{ik+j}, a_{ik+j}, g_{(i+1)k}) - \lambda_2 \mathcal{R}_i^{\pi_h, \pi_l}.$$

The training tuples for the low-level policy are formed as the per-step state-action transitions $(s_{ik+j}, g_{(i+1)k}, a_{ik+j}, r_{l,ik+j}, s_{ik+j+1}, g_{(i+1)k})$[1], which then are stored in the low-level buffer $D_l$. Thus, with the training tuples, we can optimize the low-level policy as,

$$Q^{\pi_l}(s, a) = \arg\min_Q \frac{1}{2} \mathbb{E}_{s,g,a \sim D_l} \left[ \hat{r}_l(s, a, g) + \gamma \mathbb{E}_{s',g \sim D_l, a' \sim \pi_l} Q^{\pi_l}(s', a') - Q^{\pi_l}(s, a) \right]^2,$$

$$\pi_l = \arg\min_{\pi_l} \mathbb{E}_{s,g \sim D_l} \left[ D_{KL}(\pi_l(\cdot|s, g) \| \exp(Q^{\pi_l}(s, a) - V^{\pi_l}(s))) \right].$$

**Algorithm framework.**   We briefly give an overview of our proposed BrHPO in algorithm 1. Notably, the mutual response mechanism effectively calculates the subgoal reachability for bilateral information and then incorporates it into hierarchical policy optimization for mutual error correction, promoting performance and reducing computation load.

**HIRO.**   In this work Nachum et al. (2018b), to deal with the non-stationarity, where old off-policy experience may exhibit different transitions conditioned on the same goals, they heuristically relabel the subgoal $\tilde{g}$ as

$$\log \mu^{lo}(a_{t:t+c+1}|s_{t:t+c+1}, \tilde{g}_{t:t+c+1}) \propto -\frac{1}{2} \sum_{i=t}^{t+c-1} \|a_i - \mu^{lo}(s_i, \tilde{g}_i)\|_2^2 + \text{const.}$$

To solve this problem efficiently, they calculated the quantity on eight candidate goals sampled randomly from a Gaussian centred at $s_{t+c} - s_t$. Then, with the correcting high-level experience, the high-level policy can be optimized by off-policy methods. Compared with our methods, the off-correction can be regard as a low-level domination method, which requires the high-level experience to be modified by the subgoal reachability demonstrated at low level.

---

[1]We use the absolute subgoal in this paper, that is, $g_{(i+1)k} = s_{ik} + \pi_h(\cdot|s_{ik})$.

---

**Algorithm 1** Bidirectional-reachable Hierarchical Policy Optimization (BrHPO)

---

**initialize:** policy networks $\pi_h$, $\pi_l$, $Q$-networks $Q^{\pi_h}$, $Q^{\pi_l}$, replay buffers for high-level $D_h$ and low-level $D_l$
**for** each training episode **do**
    **while** not *done* **do**
        sample subgoals $g \sim \pi_h(\cdot|s)$
        **for** each step in a subtask **do**
            Sample actions $a \sim \pi_l(\cdot|s, g)$
            Store $(s, g, a, r_l, s', g)$ into a temp buffer
            Update $\pi_l$ by (3.11) and (3.12) from $D_l$            ▷ low-level policy optimization
        **end for**
        Calculate $\mathcal{R}_i^{\pi_h, \pi_l}$ by (3.7)            ▷ subgoal reachability computation
        Compute $\hat{r}_l$ by (3.10) and push the tuples in $D_l$     ▷ reachability-aware low-level reward
        Store $(s, g, r_h, s', \hat{\mathcal{R}}_i^{\pi_h, \pi_l})$ into $D_h$
        Update $\pi_h$ by (3.8) and (3.9) from $D_l$            ▷ high-level policy optimization
    **end while**
**end for**

---

**HIGL.** In this work Kim et al. (2021), to restrict the high-level action space from the whole goal space to a $k$-step adjacent region, they introduced the shortest transition distance as a constraint in high-level policy optimization. Besides, they utilized farthest point sampling and priority queue $\mathcal{Q}$ to improve the subgoal coverage and novelty. To enhance the subgoal reachability, they made pseudo landmark be placed between the selected landmark and the current state in the goal space as follows:

$$g_t^{\text{pseudo}} := g_t^{\text{cur}} + \delta_{\text{pseudo}} \cdot \frac{g_t^{\text{sel}} - g_t^{\text{cur}}}{\|g_t^{\text{sel}} - g_t^{\text{cur}}\|^2}.$$

To establish the adjacency constraint by the shortest transition distance, they refer to HRAC Zhang et al. (2020) and adopt an adjacent matrix to model it. Specifically, we note that the performance of HIGL in AntMaze task is different from the original report in their paper, mainly due to the different scales. Thus, we set the same scale for all tasks for fairness. To ensure that HIGL performs well in these tasks, we adjusted hyper-parameters such as "landmark coverage" and "n landmark novelty".

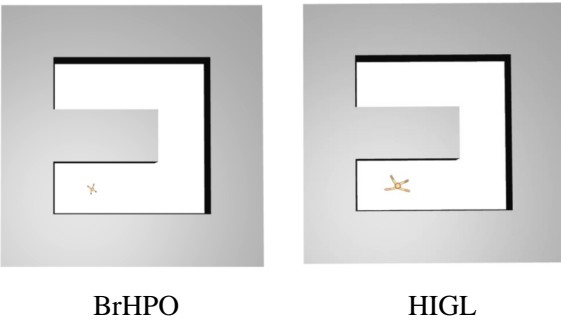

BrHPO                                   HIGL

Figure 10: Comparison of the scales in the maze tasks between BrHPO and HIGL.

**CHER.** This work Kreidieh et al. (2019) proposed a cooperation framework for HRL. In this work, the HRL problem can similarly be framed as a constrained optimization problem,

$$\max_{\pi_m} \left[ J_m + \min_{\lambda \leq 0} \left( \lambda \delta - \lambda \min_{\pi_w} J_w \right) \right].$$

To deal with this problem, they update the high- ($\pi_m$) and low-level ($\pi_w$) policies by

$$\theta_w \leftarrow \theta_w + \alpha \nabla_{\theta_w} J_w, \quad \text{and,} \quad \theta_m \leftarrow \theta_m + \alpha \nabla_{\theta_w} (J_w + \lambda J_w).$$

Compared to CHER, our BrHPO method distinguishes itself in several key aspects. In CHER, hierarchical cooperation is achieved solely through high-level policy optimization, while the low-level

policy is trained as a generally goal-conditioned policy without further improvement. Moreover, the high-level optimization in CHER introduces $J_w$ as $(J_w + \lambda J_w)$, necessitating a focus on the step-by-step behavior of the low-level policy.

In contrast, our BrHPO method incorporates the concept of subgoal reachability, which considers the initial and final states of the subtasks. This design choice empowers the high-level policy to relax the exploration burden on the low-level policy. By leveraging subgoal reachability, our approach enables more efficient exploration for the low-level policy and facilitates effective hierarchical cooperation between the high-level and low-level policies.

**RIS.**  In this work Chane-Sane et al. (2021), based on hindsight method, they collected feasible state trajectories and predicted an appropriate distribution of imagined subgoals. They first defined subgoals $s_g$ as midpoints on the path from the current state $s$ to the goal $g$, and further minimized the length of the paths from $s$ to $s_g$ and $s_g$ to $g$. Thus, the high-level policy can be updated as

$$\pi_{k+1}^H = \arg\min_{\pi^H} \mathbb{E}_{(s,g)\sim D, s_g \sim \pi^H(\cdot|s,g)}[C_\pi(s_g|s,g)].$$

Then, with the imagined subgoals, the low-level policy can be trained by

$$\pi_{\theta_{k+1}} = \arg\max_{\theta} \mathbb{E}_{(s,g)\sim D}\mathbb{E}_{a\sim\pi_\theta(\cdot|s,g)}\left[Q^\pi(s,a,g) - \alpha D_{KL}\left(\pi_\theta\|\pi_k^{prior}\right)\right].$$

### B.2  NETWORK ARCHITECTURE

For the hierarchical policy network, we employ SAC Haarnoja et al. (2018b) as both the high-level and the low-level policies. Each actor and critic network for both high level and low level consists of 3 fully connected layers with ReLU nonlinearities. The size of each hidden layer is $(256, 256)$. The output of the high- and low-level actor are activated using the linear function and is scaled to the range of corresponding action space.

We use Adam optimizer Kingma & Ba (2014) for all networks in BrHPO.

### B.3  ENVIRONMENTAL SETUP

We adopt six challenging long-term task to evaluate BrHPO, which can be categorized into the *dense* case and the *sparse* case. For the maze navigation tasks, a simulated ant starts at $(0, 0)$ and the the environment reward is defined as $r = -\sqrt{(x - g_x)^2 + (y - g_y)^2}$ (except for AntFall, $r = -\sqrt{(x - g_x)^2 + (y - g_y)^2 + (z - g_z)^2}$). While in the robotics manipulation tasks, a manipulator is initialized with horizontal stretch posture. The environmental reward is defined as a binary one, determined by the distance between the end-effector (or the object in Pusher) to the target point

$$r = \begin{cases} -1, & d > 0.25, \\ 0, & d \le 0.25. \end{cases} \tag{B.1}$$

And, the success indicator is defined as whether the final distance is less than a pre-defined threshold, where the maze navigation tasks require $d < 5$ and the robotics manipulation tasks require $d < 0.25$.

**AntMaze.**  A simulated eight-DOF ant starts from the left bottom $(0, 0)$ and needs to approach the left top corner $(0, 16)$. At each training episode, a target position is sampled uniformly at random from $g_x \sim [-4, 20], g_y \sim [-4, 20]$. At the test episode, the target point are fixed at $(g_x, g_y) = (0, 16)$.

**AntBigMaze.**  Similar to AntMaze task, we design a big maze to evaluate the exploration capability of BrHPO. Specially, the target position is chosen randomly from one of $(g_x, g_y) = (32, 8)$ and $(g_x, g_y) = (66, 0)$, which makes it harder to find feasible path.

**AntPush.**  A movable block at $(0, 8)$ is added into this task. The ant needs to move to the left side of the block and push it into the right side of the room, for a chance to reach the target point above, which requires the agent to avoid training a greedy algorithm. At each episode, the target position is fixed to $(g_x, g_y) = (0, 19)$.

**AntFall.** In this task, the agent is initialized on a platform of height 4. Like AntPush environment, the ant has to push a movable block at $(8, 8)$ into a chasm to create a feasible road to the target, which is at the opposite side of the chasm, while a greedy policy would cause the ant to walk towards target and fall into the chasm. At each episode, the target position is fixed to $(g_x, g_y, g_z) = (0, 27, 4.5)$.

**Reacher3D.** A simulated 7-DOF robot manipulator needs to move its end-effector to a desired position. The initial position of the end-effector is at $(0, 0, 0)$ while the target is sampled from a Normal distribution with zero mean and 0.1 standard deviation.

**Pusher.** Pusher additionally includes a puck-shaped object based on the Reacher3D task, and the end-effector needs to find the object and push it to a desired position. At the initialization, the object is placed randomly and the target is fixed at $(g_x, g_y, g_z) = (0.45, -0.05, -0.323)$.

We summary these six tasks in Table 1.

Table 1: Overview on Environment settings.

| Environment | state | action | environment reward | episode step | success indicator |
|:---:|:---:|:---:|:---:|:---:|:---:|
| AntMaze | 32 | 8 | negative x-y distance | 500 | $r_{\text{final}} \geq -5$ |
| AntBigMaze | 32 | 8 | negative x-y distance | 1000 | $r_{\text{final}} \geq -5$ |
| AntPush | 32 | 8 | negative x-y distance | 500 | $r_{\text{final}} \geq -5$ |
| AntFall | 33 | 8 | negative x-y-z distance | 500 | $r_{\text{final}} \geq -5$ |
| Reacher3D | 20 | 7 | negative x-y-z distance | 100 | $d_{\text{final}} \leq 0.25$ |
| Pusher | 23 | 7 | negative x-y-z distance | 100 | $d_{\text{final}} \leq 0.25$ |

## B.4 HYPER-PARAMETERS

Table 2 lists the hyper-parameters used in training BrHPO over all tasks.

Table 2: The hyper-parameters settings for BrHPO.

| | AntMaze | AntBigMaze | AntPush | AntFall | Reacher3D | Pusher |
|---|---|---|---|---|---|---|
| $Q$-value network (both high and low) | MLP with hidden size 256 | | | | | |
| policy network (both high and low) | Gaussian MLP with hidden size 256 | | | | | |
| discounted factor $\gamma$ | 0.99 | | | | | |
| soft update factor $\tau$ | 0.005 | | | | | |
| $Q$-network learning rate | 0.001 | | | | | |
| policy network learning rate | 0.0001 | | | | | |
| automatic entropy tuning (high-level) | False | | True | | False | |
| automatic entropy tuning (low-level) | False | | | | | |
| batch size | 128 | | | | | |
| update per step | 1 | | | | | |
| target update interval | 2 | | | | | |
| high-level replay buffer | 1e5 | | | | | |
| low-level replay buffer | 1e6 | | | | | |
| start steps | 5e3 | | | | | |
| subtask horizon | 20 | | | | 10 | |
| reward scale | 1 | | | | | |
| high-level responsive factor $\lambda_1$ | 2 | | 0.5 | | 2 | |
| low-level responsive factor $\lambda_2$ | 10 | | | | 5 | |

## B.5 Additional experiments

**Mutual response mechanism in complex environment.** In our main experiment, AntBigMaze uses a more complex maze than AntMaze, which requires more guidance ability from the high-level policy. Conversely, given simpler maze, we consider a more complex robot which is controlled by the low-level policy, to further evaluate the mutual response mechanism. In the HumanoidMaze task, the simulated ant is replaced by a high-dimensional simulated humanoid. At each episode, the target point is set at $(g_x, g_y) = [6, 6]$ which is the up-right corner of the maze.

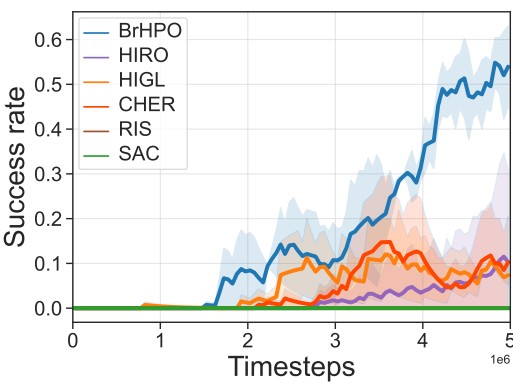

Figure 11: The performance comparison between BrHPO and baselines by HumanoidMaze. Mean and std by 4 runs.

We need to specify that, the simulated humanoid, where the state space contains 274 dimensions and the action space is 17, needs to maintain body balance while being guided by the subgoal from the high-level policy. Consequently, the low-level policy necessitates extensive training to facilitate the humanoid's ability to learn how to walk proficiently. This training process requires the high-level policy to exhibit "patience", gradually adjusting the subgoals to guide the humanoid's progress effectively. Figure 11 demonstrates the performance comparison, which showcases the superior advantage of BrHPO over HIRO. We additionally visualize the trajectory by Figure 12. We find that, our mutual response mechanism can encourage cooperation between the high- and the low-level policies, while the erroneous guidance from HIRO makes it difficult for humanoid to maintain balance and easily fall, thus failing the task.

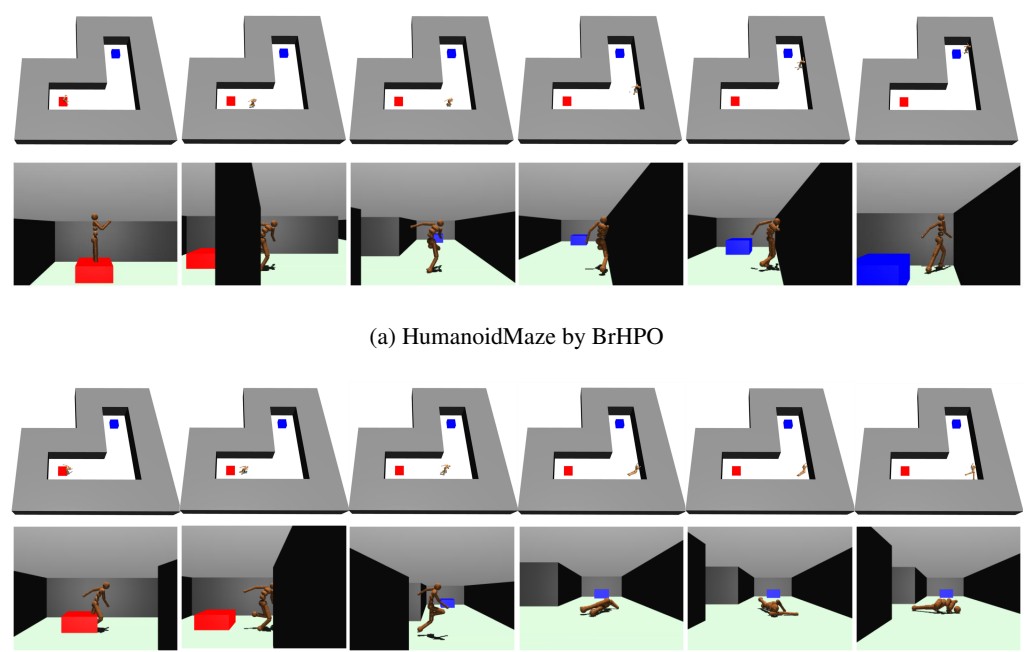

(a) HumanoidMaze by BrHPO

(b) HumanoidMaze by HIRO

Figure 12: The performance comparison of HumanoidMaze task by BrHPO and HIRO.

**Additional Metrics.** We report additional (aggregate) performance metrics of BrHPO and other baselines on the six tasks using the `rliable` toolkit Agarwal et al. (2021). As show in Figure 13, BrHPO outperforms other baselines in terms of Median, interquantile mean (IQM), Mean and Optimality Gap results.

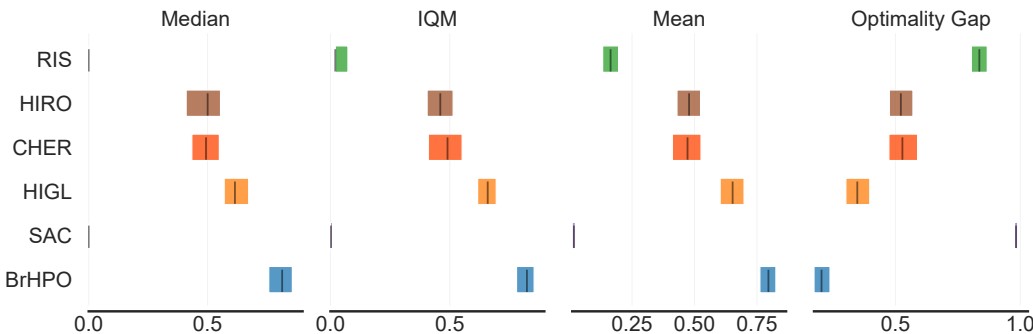

Figure 13: Median, IQM, Mean (higher values are better) and Optimality Gap (lower values are better) performance of BrHPO and all baselines on six tasks.

**Subgoal reachability report.** We report the average subgoal reachability $\mathcal{R}_i^{\pi_h, \pi_l}$ of each environment by Table 3. Note that, the value $\mathcal{R}_i^{\pi_h, \pi_l} \to 0$ means the final distance $\mathcal{D}(\psi(s_{(i+1)k}), g_{(i+1)k}) \to 0$, thus implying the better subgoal reachability. From the results, our implementation is simple yet effective, which can improve subgoal reachability significantly. Besides, the results shows that when there are contact dynamics in the environment, such as AntPush, AntFall and Pusher, the subgoal reachability may be decreased, which inspires us to further develop investigation in these cases.

Table 3: The average subgoal reachability of BrHPO.

| Environment | AntMaze | AntBigMaze | AntPush | AntFall | Reacher3D | Pusher |
|---|---|---|---|---|---|---|
| subgoal reachability | 0.22 | 0.29 | 0.33 | 0.32 | 0.13 | 0.18 |

**Ablation by the sparse environment.** Additionally, we provide ablation studies conducted on Reacher3D task (*sparse*) instead of the AntMaze task (*dense*). We investigate the effectiveness of mutual response mechanism by 1) the three variants of BrHPO, containing ***Vanilla***, ***NoReg*** and ***NoBonus***, and 2) the weighted factors $\lambda_1$ and $\lambda_2$. We show the results in Figure 14. Overall, we find that the tendency from the Reacher3D task are similar to the AntMaze task, which verifies the effectiveness of our BrHPO in the *sparse* reward case.

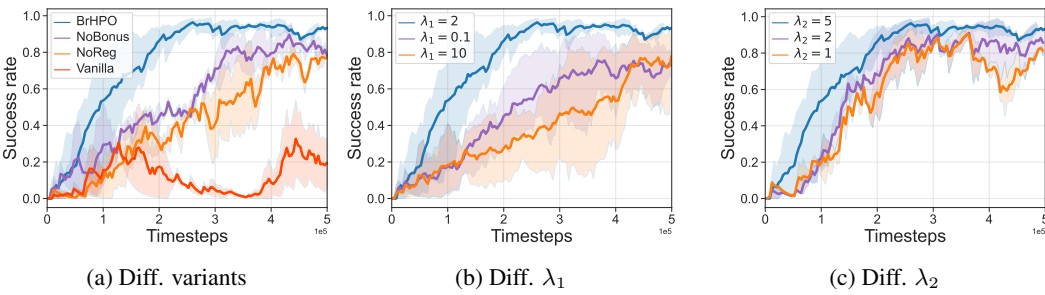

(a) Diff. variants  (b) Diff. $\lambda_1$  (c) Diff. $\lambda_2$

Figure 14: The ablation of mutual response mechanism by Reacher3D task. Mean and std by 4 runs.

**Empirical study in stochastic environments.** To empirically verify the stochasticity robustness of BrHPO, we utilize it the a set of stochastic tasks, including stochastic AntMaze, AntPush and Reacher3D, which are modified from the original tasks. Referring to HRAC Zhang et al. (2020), we interfere with the position of the ant (x,y) and the position of the end-effector (x,y,z) with Gaussian noise of different standard deviations, including $\sigma = 0.01$, $\sigma = 0.05$ and $\sigma = 0.1$, to verify the robustness against the increasing environmental stochasticity. As shown in Figure 15, BrHPO can achieve similar asymptotic performance with different noise magnitudes in stochastic AntMaze, AntPush and Reacher3D, which shows the robustness to stochastic environments.

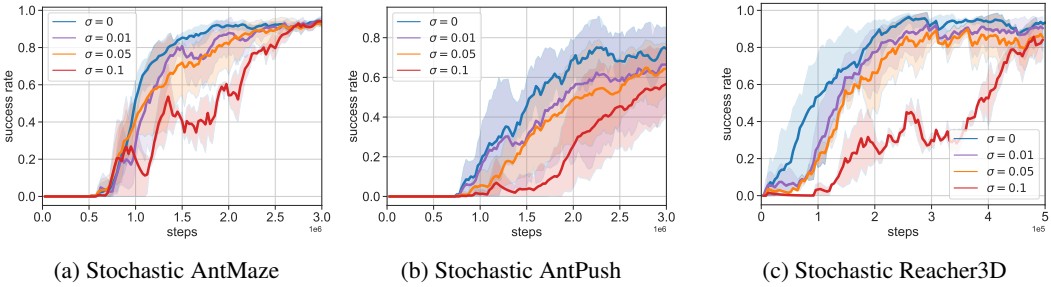

| (a) Stochastic AntMaze | (b) Stochastic AntPush | (c) Stochastic Reacher3D |

Figure 15: The empirical evaluation of BrHPO by stochastic environments. Mean and std by 4 runs.

## B.6 COMPUTING INFRASTRUCTURE AND TRAINING TIME

For completeness, we list the computing infrastructure and benchmark training times for BrHPO and all baselines by Table 4. As discussed in section 4.3, the training complexity of BrHPO is much less than other HRL methods, which can be comparable to the flat policy.

Table 4: Computing infrastructure and training time on each task (in hours).

|  | AntMaze | AntBigMaze | AntPush | AntFall | Reacher3D | Pusher |
|---|---|---|---|---|---|---|
| CPU | AMD EPYC™ 7763 | | | | | |
| GPU | NVIDIA GeForce RTX 3090 | | | | | |
| HIRO | 16.66 | 23.14 | 18.29 | 25.43 | 3.42 | 4.25 |
| HIGL | 31.59 | 48.45 | 30.95 | 49.60 | 5.96 | 7.05 |
| CHER | 15.38 | 20.53 | 16.71 | 21.37 | 2.96 | 3.16 |
| RIS | 40.83 | 53.49 | 38.46 | 57.05 | 8.63 | 9.88 |
| **SAC** | **10.57** | **11.36** | **11.75** | **15.64** | **2.35** | **2.68** |
| **BrHPO** | **12.75** | **18.74** | **13.43** | **19.17** | **2.73** | **3.53** |
| **comparison (Ours - SAC)** | **2.18** | **7.38** | **1.68** | **3.53** | **0.38** | **0.85** |

