# OpenReview forum: "Bidirectional-Reachable Hierarchical RL with Mutually Responsive Policies"
_ICLR.cc/2024/Conference — Submitted to ICLR 2024_

### Official Review · Reviewer_QVuZ · 2023-10-30

**Soundness:** 1 poor
**Presentation:** 3 good
**Contribution:** 2 fair
**Rating:** 5
**Confidence:** 3

**Summary:**

The paper discusses subgoal reachability as a key issue in subgoal-based hierarchical reinforcement learning (HRL) and argues that existing methods tackle this problem in a one-sided way by making one level dominant. As this can lead to shortcomings like poor exploration, the authors propose a method for improving subgoal reachability that tries to strike a balance between the levels. At its core is a subgoal reachability metric that is defined by the distance to the subgoal at the end of a subtask divided by the distance at the beginning of the subtask. By subtracting this metric as a regularization term on the higher level and an auxiliary reward term on the lower level, subgoal reachability can be improved and training stabilized. A bound for the sub-optimality of the hierarchy and experiments on HRL-typical environments are presented where the latter show that the proposed method, BrHPO, outperforms a range of HRL baselines.

**Strengths:**

The paper correctly identifies effective communication between the levels of a hierarchy as one of the cornerstones of successful HRL. The discussion of the shortcomings of existing methods is an adequate motivation for the proposal of a bidirectional method.

The proposed method is furthermore simple to integrate into existing subgoal-based HRL algorithms and computationally inexpensive.

The empirical results show that the regularization and reward terms speed up and stabilize training. The ablation studies adequately demonstrate that the regularization of the high-level policy results in the biggest improvement while the auxiliary reward term on the lower level has a smaller positive effect on training.

**Weaknesses:**

Theorem 3.2 has too many problems. First, the assumptions should be stated clearly in the Theorem itself. Instead, Assumption A.5 is only mentioned in the proof above equation (A.11). Assumption A.5 is furthermore not stated properly. On the RHS variables $s_t$, $a_t$ show up that have not been defined. Furthermore, the RHS does not depend on $s$ and $a$ anymore at all because $s$ and $a$ are taken to be distributed according to $\pi_l$ on the RHS. What is written above (A.11) is different again. Now $s$, $a$ do not enter the expression in the expectation value anymore. An even bigger problem is that Assumption A.5 requires the environment reward to depend on the low-level policy. In the standard RL setting this is not possible and Assumption A.5 is not even satisfied in the environments considered in the experiments. This reward is furthermore not bounded as setting $g$ close to $\hat{g}$ will make it arbitrarily large, contrary to the implicit assumption that the reward is bounded.

The problems continue in the line after (A.11). There, the claim is made that $r_l(s_j, a_j, g) \leq r_l(s_k, a_k, g)$ for $0\leq j \leq k$. This is wrong, however, because $\pi_l$ is not even optimal, and even if it were optimal, this might not hold in some environments. Below (A.13) there is another claim that is not proven, i.e., $\mathcal{R}^{\pi_h, \pi_l} \geq \mathcal{R}^{\pi_h^*, \pi_l^*} $, and that is not self-evident.
There might be other problems in the parts of the proof I did not read but these issues alone are enough to make the proof incorrect. The bound in Theorem 3.2 is furthermore not tight at all because of a constant term $4r_\text{max}\gamma^k/(1-\gamma)^2$. Because of $|V(s_0)| \leq r_\text{max}/(1-\gamma)$ this term alone is bigger than a naive bound $C(\pi_h, \pi_l) \leq 2r_\text{max}/(1-\gamma)$, assuming $r \in [-r_\text{max}, r_\text{max}]$. It is therefore questionable if the bound was very useful, even if it was correct.

Another big issue with the paper is that Hierarchical Actor-Critic (HAC) [1] is not discussed (but only cited in passing) even though it is highly relevant. In HAC, the higher level receives a constant penalty if the lower level cannot reach its assigned subgoal. This feedback mechanism is quite similar to the regularization on the higher level of BrHPO. To see this, recall that HAC uses a shortest path objective so the reward is -1 if the goal is not reached and 0 otherwise. In the BrHPO formalism, this can be expressed via $\mathcal{D}(s, g) =\mathbb{1}_{d(s,g)> \epsilon}$ where $d$ is some distance metric. Then the subgoal reachability regularization term vanishes when the subgoal is reached in a subtask and is constant otherwise, which is equivalent to the HAC case. This similarity should be discussed and it is also in conflict with the claim that other methods do not have a feedback mechanism across the levels of the hierarchy. HAC is furthermore also tackling the issue of an effective communication between the levels with its own set of methods. It should therefore be considered as a baseline.

The higher level of BrHPO is trained with the off-policy algorithm SAC. This is problematic because the lower level changes during training and the experience in the replay buffer of the higher level therefore becomes outdated. Other algorithms like HIRO or HAC have developed methods to deal with this non-stationarity and enable efficient off-policy training. It should therefore be discussed why BrHPO does not need such methods.

In the experiments section I am missing which distance measure $\mathcal{D}$ and which mapping $\Psi$ was used.

In the paragraph “Computation load” a performance guarantee is mentioned. It is not clear to me what is meant by that.

In summary, due to the many problems with the theoretical analysis and the lack of a discussion of the similarities with HAC, I cannot recommend the paper for acceptance in its current form.

[1] Andrew Levy, George Konidaris, Robert Platt, and Kate Saenko. Learning multi-level hierarchies with hindsight. In International Conference on Learning Representations, 2019.

**Questions:**

* It looks like $\hat{g}$ is not really introduced in the main text, is that correct?
* The definition of the transition probabilities looks like the state space was chosen as discrete. Is this intentional?

---

### Official Review · Reviewer_VKLy · 2023-11-01

**Soundness:** 3 good
**Presentation:** 3 good
**Contribution:** 2 fair
**Rating:** 5
**Confidence:** 3

**Summary:**

The authors present a new setup for hierarchical reinforcement learning. They argue that reachable subgoals can be better generated through better usage of information between high-level and low-level planning. They evaluate their approach in an extensive empirical study and show improvement over other state-of-the-art HRL approaches.

**Strengths:**

The paper starts with a good motivating example, makes an effort to explain its approach very precisely, and a good array of example cases. It includes an ablation study and thus makes some effort to substantiate its claims.

**Weaknesses:**

Between the multitude of information the paper presents, it clearly loses focus of the story it -- in some parts still visibly -- wants to tell. Most equations should be simplified drastically and re-writings should be cut in favor of presenting the main idea. Theorem 3.2 (which should probably appear as Theorem 3.1?) appears not impactful in this context (and if it is, it should probably be a paper on its own).

The evaluation study, while extensive, never discusses the most important aspects: Do the benefits come at a cost? Are there instances where the additional aspects of the algorithm would hinder progress? What does this say about the usability of any other HRL approach? If the right answers could be given to these questions, the impact of the paper would be improved dramatically.

As of now, the paper remains unclear if it describes an intensive parametrization process for a few select domains or a general addition to HRL and somehow wants to present its result as both.

The "extension in complex environment" is in this form not helpful and should be presented clearer (with more space) or simply cut from the paper.

Furthermore, the comparison to vanilla SAC certainly feels off. What happens in domains where vanilla apporaches perform better than "laughable"?

Minor notes:
- Use "behavior" not "behaviour" since you use American English thorughout
- Write "It is" and "cannot" instead of "It's" and "cannot"
- For me, Equation 2.1 should use \langle and \rangle instead of \{ and \}
- Just below Equation 2.3, the type of \psi should use \to not \mapsto

**Questions:**

see above

---

### Official Review · Reviewer_cpCG · 2023-11-01

**Soundness:** 2 fair
**Presentation:** 3 good
**Contribution:** 3 good
**Rating:** 6
**Confidence:** 4

**Summary:**

This work studies Hierarchical Reinforcement Learning (HRL), pointing out that existing methods only consider unilateral subgoal reachability and ignore the significance of bilateral subgoal reachability. Starting from theoretical analysis, this paper presents an upper bound of suboptimal performance difference characterized by a reachability metric. Then this paper proposes a new HRL algorithm called Bidirectional-reachable Hierarchical Policy Optimization (BrHPO) is proposed. BrHPO uses the subgoal reachability metric as policy optimization regularizer and additional bonus for high-level and low-level learning respectively. The proposed methods are evaluated against representative HRL baselines in several continuous control tasks with both dense and sparse reward. Diverse experiments on ablations and other analysis are provided.

**Strengths:**

- The paper is overall well organized and written.
- The theoretical derivation is almost clear and easy to follow. The appendix provides sufficient details.
- The presentation of the proposed algorithm is clear. The proposed algorithm is simple and easy to implement with significant better performance than considered baselines and friendly training cost.
- The experiments are conducted from diverse aspects.

**Weaknesses:**

- I feel that the meaning of ‘bilateral reachability’ is not clear enough (also for the associated concepts like ‘Low-level dominance’ and ‘High-level dominance’). Concretely, to me, I have no problem with the high-level dominance as a few recent HRL works focus on this point, while the low-level dominance is not clear.
- The transition from Section 1,2 and Section 3 is not smooth. I do not see how the theory introduced in Section 3 closely connects to the ‘unilateral-bilateral reachability’ problem.
- The derivation of the main theory seems to include some assumptions (in the appendix) about which I think more discussion and explanation are needed.
- Some details are the proposed algorithm is not clear (please see the questions below).

**Questions:**

1) Is the bonus term in Equation 3.10 computed in a hindsight manner, as it can only be computed at the timestep $(i+1)k$ rather than $ik + j$? If so, how to consider the off-policyness (i.e., the low-level policy changes) in the bonus term when doing experience replay?
2) According to Equation 3.8 and 3.7, I do not see how the regularization term can be differentiable with respect to the policy (parameter). Can the authors explain on this?
3) It is not straightforward to understand the term in Equation 3.5. After checking the appendix, I found it seems that a few assumptions are made, e.g., ‘Since the low-level policy is trained as a goal-conditioned policy’, ‘consider that the subgoals are generated towards the environmental goal, when given a low-level optimal/learned policy’. I think this deserves more discussion in the main body of this paper.

---

### Meta-Review · Area_Chair_oCq6 · 2023-12-14

**Metareview:**

This paper revise the idea of what the author call bilateral information sharing, propose a theoretical framework, and finally a practical implementation of this idea.

Overall, the idea is neat, and the proposed algorithm seem to yield good empirical results. However, the reviewers pointed out important concerns. In particular, I here highlight how all three reviewers have comments regarding the theoretical proofs. Moreover, Reviewer QVuZ raised a legitimate point regarding the similarities with the HAC algorithm, and how this should be included as a baseline in the experiments.

Unfortunately, the authors decided not to participate in the rebuttal phase, thus leaving the concerns of the reviewers unanswered.

**Justification For Why Not Higher Score:**

The authors did not participate in the rebuttal phase.

**Justification For Why Not Lower Score:**

N/A

---

### Decision · Program_Chairs · 2024-01-16

Reject